# Spectral Bias in Practice: the Role of Function Frequency in Generalization

## Abstract

Despite their ability to represent highly expressive functions, deep learning models trained with SGD seem to find simple solutions that generalize surprisingly well. Spectral bias – the tendency of neural networks to prioritize learning low frequency functions – is one possible explanation for this phenomenon, but so far spectral bias has primarily been observed in theoretical models and simplified experiments. In this work, we propose methodologies for measuring spectral bias in modern image classification networks. We find that these networks indeed exhibit spectral bias, and that interventions that improve generalization sometimes increase and sometimes decrease the frequencies of the learned function. For example, we experimentally show that larger models learn high frequencies more readily than smaller ones, but many forms of regularization, both explicit and implicit, inhibit the learning of high frequencies. We also explore the connections between function frequency and image frequency and find that spectral bias is sensitive to the low frequencies prevalent in natural images. Our work enables measuring and ultimately influencing the spectral behavior of neural networks used for image classification, and is a step towards understanding why deep models generalize well.

## 1 Introduction

Two fundamental questions in machine learning are why overparameterized models generalize and how to make them more robust to distribution shift. Resolving both of these questions requires understanding how complex our models should be.

For instance, it is thought that overparameterized models generalize well because there are implicit regularizers that constrain the complexity of the learned functions. However, the precise nature of these implicit regularizers, and their importance in practice, remains unclear. As for how to achieve robustness, no consensus has emerged regarding function complexity. On the one hand, Cranko et al. (2018) and Leino et al. (2021) argue that Lipschitz smoothness of the learned function offers a guarantee of robustness; on the other, Shah et al. (2020) and Madry et al. (2019) argue that simplicity can be harmful and a robust model must actually be more complex than its non-robust counterpart.

One window into function complexity is spectral bias – the tendency of neural networks to learn low frequency (simple and smooth) functions early in training, gradually increasing the frequency (complexity) of the learned function as training proceeds. Foundational work in this area has shown theoretical evidence of spectral bias by analyzing convergence rates of neural networks towards functions of different frequencies and showing that they converge to low frequency functions at a faster rate (Basri et al., 2019; Rahaman et al., 2019).

In practice, spectral bias is difficult to measure: the most direct method involves taking a Fourier transform with respect to the input, which is expensive to compute due to the high dimensionality of images. Early experimental work focused on low-dimensional synthetic data (Basri et al., 2019; 2020) or used proxy measurements of spectral bias by inserting label noise of various frequencies during training (Rahaman et al., 2019). However, the label noise method is limited to binary classification/regression and involves modifying the training data, making it difficult to disentangle how other changes to the training data (such as data augmentation) affect the spectral content of the learned function. Thus, it remains an open question as to whether modern neural networks exhibit spectral bias and what role it plays in generalization.

In this work, we investigate model complexity through the lens of spectral bias, by introducing experimental methods to study the frequency decomposition of the functions learned by modern image classification networks. We find that neither simplicity nor complexity is purely beneficial or detrimental to performance.

**Contributions.** We extend the label noise procedure of Rahaman et al. (2019) to enable measuring spectral bias in multi-class classification and apply this technique to understand the function frequencies present in high-accuracy models on CIFAR-10 (Krizhevsky et al.). We also introduce a second method for measuring the smoothness of a learned function via linear interpolation between test examples. This offers a proxy measurement of spectral bias without the need to modify training labels, allowing us to probe the effects of the training data on the learned function frequencies.

Using these experimental methods, we find that increasing the model size decreases spectral bias; larger models learn high frequencies faster. However, we observe that more accurate models are not always higher frequency. Several forms of explicit and implicit regularization, including weight decay, increasing the dataset size, and applying Mixup (Zhang et al., 2018) data augmentation, increase spectral bias and produce a lower-frequency learned function. Our experiments show that these common training interventions affect the frequency content of the learned function in different ways, helping us to understand their mechanism and offering a tool to aid design of improved training methods in the future.

This perspective allows us to shed light on the mechanism behind self-distillation, in which a student model is trained to fit the predictions of a teacher model. Using our linear interpolation methodology, we observe that self-distillation produces a student model whose learned function is smoother than that of its teacher. This suggests that the teacher model acts as a sort of low-pass filter on the target function, perhaps making it more accessible to the spectrally-biased student.

Finally, we explore the relationship between image frequency and function frequency. By further extending the label noise methodology of Rahaman et al. (2019) to study spectral bias in directions of interest through the input space, we find that models are most sensitive to the low image frequencies common in natural images.

## 2 RELATED WORK

**Implicit bias.** A common belief is that some form of implicit bias imposed by the training procedure may account for the generalization ability of overparameterized neural networks, and accordingly, much research has been directed towards understanding these implicit biases (Soudry et al., 2018; Zhang et al., 2017; Keskar et al., 2017; Gunasekar et al., 2019; 2020; Hardt et al., 2016; Hoffer et al., 2018; Belkin et al., 2018; Neyshabur et al., 2018; 2015; 2017; Mania et al., 2019; Gunasekar et al., 2019; Nakkiran et al., 2019). One of the most well-known forms of implicit bias comes from the optimization algorithm itself, which typically guides networks to solutions that generalize well (Soudry et al., 2018; Keskar et al., 2017; Wilson et al., 2017).

**Spectral bias.** Spectral bias is a form of implicit bias, with much recent attention including theoretical and experimental approaches (Oymak et al., 2019; Xu et al., 2019; Rahaman et al., 2019; Basri et al., 2019; Zhang et al., 2021); we describe some of these prior results here. Basri et al. (2019) studied spectral bias using a linear model of SGD training dynamics to show that models learn low frequency (simple) functions early in training and then gradually learn higher frequencies as training proceeds, assuming training data that is distributed uniformly on the hypersphere. Basri et al. (2020) extended the analysis to consider nonuniformly spaced training data, and found that learning is faster where samples are denser; if sampling is nonuniform, then during training, the learned function will be higher frequency in regions with denser samples.

Rahaman et al. (2019) used a more direct analysis to show the same spectral bias toward low frequency functions and also suggested that when training data lie on a manifold (as natural images are believed to), increasing the complexity of the manifold makes it easier for models to fit high frequency functions. Additionally, Rahaman et al. (2019) posited that high frequency components of the learned function are most sensitive to perturbations in the model parameters, connecting back to the idea of flat optimization minima (Neyshabur et al., 2017) or smooth loss landscape Mehmeti-Göpel et al. (2021). They proposed experimental methods to study spectral bias in image classification, but

focused on a binary subset of the relatively simple MNIST dataset (Deng, 2012), with mean square error loss.

Our label smoothing experiments are a direct extension of Rahaman et al. (2019) to multiclass classification with the more common cross-entropy loss. Our linear interpolation experiments are somewhat similar to the recently-proposed experimental methods in Zhang et al. (2021), except that we sample along paths between images rather than in regions surrounding each image; this allows for a more global measurement of spectral bias.

**Model sensitivity to image frequency.** While we focus on *function frequency* in this work, prior research aimed to understand model sensitivity to image frequencies. Jo & Bengio (2017) found that CNNs are sensitive to Fourier statistics of the training data, even those irrelevant to human viewers. Ortiz-Jimenez et al. (2020b) studied the image frequency bias induced by using a convolutional architecture, and Ortiz-Jimenez et al. (2020a) argued that models are sensitive primarily to discriminative Fourier directions in the training data. They posit that adversarial training induces large margins in orthogonal directions, thereby improving robustness. Yin et al. (2020) introduced a procedure to measure the sensitivity of trained models to Fourier image perturbations, and showed that adversarial training and Gaussian data augmentation increase robustness to high frequency perturbations but increase sensitivity to low frequency perturbations. Our work adds to this line of research a study of the interplay between two notions of frequency: the function frequency involved in spectral bias, and the image frequencies present in the data.

## 3 METHODOLOGY

The goal of our work is to measure the complexity of neural network functions through the lens of frequency. In low dimensions, measuring the frequency decomposition of a function is straightforward and tractable: evaluate the function at dense, uniform sampling positions and compute its discrete Fourier transform. However, the function of interest in image classification is unavoidably high-dimensional, mapping images (with thousands of pixel values) to object classes. It would be intractable even to collect sufficient samples of this function to compute a discrete Fourier transform, let alone compute the transform itself. Instead, we employ two complementary approaches to measure informative proxies of this frequency decomposition.

### 3.1 LABEL SMOOTHING

The core idea for measuring function frequency introduced by Rahaman et al. (2019) is to construct a sinusoid over the space of images, and to use that sinusoid as a form of label noise during training. Let $\mathcal{D} = \{(\mathbf{X}_i, \boldsymbol{y}_i)\}_{i=1}^{n_{train}}$ be the training examples and $\mathcal{D}_{\text{valid}} = \{(\mathbf{X}_j, \boldsymbol{y}_j)\}_{j=1}^{n_{val}}$ be the validation examples, with $\mathbf{X}_i$ an image and $\boldsymbol{y}_i$ a one-hot class encoding, where $n_{train}$ is the number of training examples, $n_{val}$ is the number of validation examples, $d$ is the side length of an image (assumed to be square for simplicity), $c$ is the number of color channels, and $M$ is the number of classes, so $\mathbf{X}_i \in \mathbb{R}^{d \times d \times c}$ and $\boldsymbol{y}_i \in \mathbb{R}^M$.

To extend this procedure to the multi-class setting, we add noise of various frequencies to an $M$-dimensional label vector via label smoothing (Szegedy et al., 2015). Let $S : \mathbb{R}^{d \times d \times c} \to [0, 1]$ be a target function that maps an input image $\mathbf{X}_i$ to a scalar value between 0 and 1. We apply label smoothing to each label $\boldsymbol{y}_i$, mapping it to $\bar{\boldsymbol{y}}_i = \boldsymbol{y}_i(1 - S(\mathbf{X}_i)) + \frac{1}{M}S(\mathbf{X}_i)$ to retain a valid probability distribution. We then train from scratch using the original examples $\mathbf{X}_i$ and their smoothed labels $\bar{\boldsymbol{y}}_i$. Finally, we evaluate on the validation images $\mathbf{X}_j$, comparing to both their original one-hot labels $\boldsymbol{y}_j$ and smoothed labels $\bar{\boldsymbol{y}}_j = \boldsymbol{y}_j(1 - S(\mathbf{X}_j)) + \frac{1}{M}S(\mathbf{X}_j)$. Our experiments use the CIFAR-10 dataset (Krizhevsky et al.), where $n_{train} = 50000$, $n_{val} = 10000$, $d = 32$, $c = 3$, and $M = 10$; however, our experimental setup is sufficiently general as to transfer easily to any image classification task.

This method is introduced in Figure 1. As training proceeds, we see both the training loss (between predictions and smoothed labels $\bar{\boldsymbol{y}}_i$) and noisy validation loss (between predictions and smoothed labels $\bar{\boldsymbol{y}}_j$) decrease. The clean validation loss (between predictions and one-hot labels $\boldsymbol{y}_j$), however, initially decreases but at some point in training plateaus or begins to increase. At this point, the model has begun to learn the target function $S$. Accordingly, we introduce **noise fitting**, defined as the difference between the validation loss on one-hot labels and the validation loss on labels smoothed with the same function $S$ that was applied to the training labels:

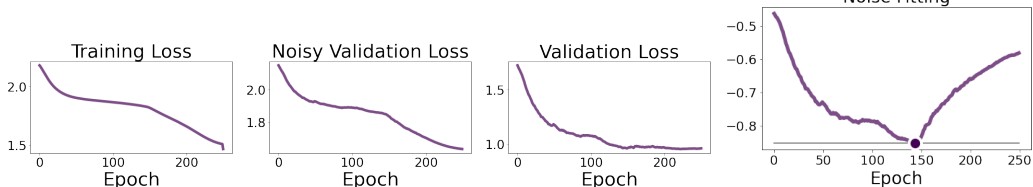

Figure 1: **Noise fitting shows when and how much a network fits a target frequency function.** At roughly epoch 150, noise fitting *(Right)*, the difference between clean and noisy validation loss, exhibits a clear "dip" when the model begins to fit the target function: training *(Left)* and validation *(Center Left)* loss on the perturbed function drop, while improvement stalls on clean validation data *(Center Right)*. By comparing the minimum value of noise fitting achieved by different models throughout training (illustrated by the gray line in the right plot), we can compare the relative degree to which different models fit noise functions of varying frequency. A model with higher **min noise fitting** more readily fits the target noise function. Here, we train a $\mathrm{wrn\_32}$ model (wide-resnet with width 32) with radial wave label smoothing at frequency 0.04. For visual clarity, we apply exponential averaging to all curves.

$$\textbf{noise fitting} = \textbf{clean validation loss} - \textbf{noisy validation loss} \tag{1}$$

We compare different models by computing the minimum noise fitting achieved throughout training. A model with higher **min noise fitting** more readily fits the target function $S$ than a model with lower min noise fitting.

By choosing different functions $S$, we can probe nuances of spectral bias. Typically (inspired by Rahaman et al. (2019)) we choose a radial wave: $S(\mathbf{X}) = \sin(2\pi f(\|\mathbf{X}\| - \mathbb{E}_{\mathcal{D}}\|\mathbf{X}\|))$, where we can vary the frequency $f$ to understand spectral bias at this global scale. We can also choose more targeted functions $S$; for instance, if $\mathbf{V}$ is a direction of interest through the space of images (*i.e.* $\mathbf{V}$ is an image-shaped vector of unit norm), we can construct $S(\mathbf{X}) = \sin(2\pi f \langle \mathbf{X}, \mathbf{V} \rangle)$. This allows us to vary both the frequency and direction of the target sinusoid, to understand model sensitivity along different directions through image space.

However, label smoothing has a few limitations. It requires retraining each model from scratch, with substantial investment of time and computational resources. Because it involves perturbing the training labels, it cannot directly answer questions about the interaction between the training dataset and spectral bias, such as the effects of data augmentation. For the same reason, we cannot use this methodology to study the spectral decomposition of existing trained models.

## 3.2 LINEAR INTERPOLATION

To complement the label smoothing approach, we also propose a measurement methodology based on linear interpolation between validation images. Although we cannot take dense, regularly-spaced samples that cover the entire input (image) space, we can sample along specific paths through, and thereby glean glimpses into the spectral content of the learned function.

We consider two types of paths: those between images from the same class, and those between images from different classes. When considering paths between images in the same class, we choose 200 random, distinct pairs of images from each of the 10 CIFAR-10 classes and average over the paths defined by these pairs. When interpolating between classes, we choose 200 random, distinct image pairs from each of the 45 (10 choose 2) pairs of distinct classes, and average over the resulting paths.

For each pair of images $(\mathbf{X}_0, \mathbf{X}_1)$, we vary $\lambda \in [0, 1]$ to trace out a path, where each image in the path is given by $\mathbf{X}_\lambda = \lambda \mathbf{X}_1 + (1 - \lambda)\mathbf{X}_0$. We choose $\lambda$s so that the distance between adjacent images on the path is constant, which means that each path may produce a different number of samples depending on the total distance between $\mathbf{X}_0$ and $\mathbf{X}_1$. Before interpolating, we normalize $\mathbf{X}_0$ and $\mathbf{X}_1$. Figure 2 *(Bottom Left)* shows an illustrative example of two images and a corresponding interpolation path between them. For our experiments, we typically have 50 to 100 images along the interpolation path.

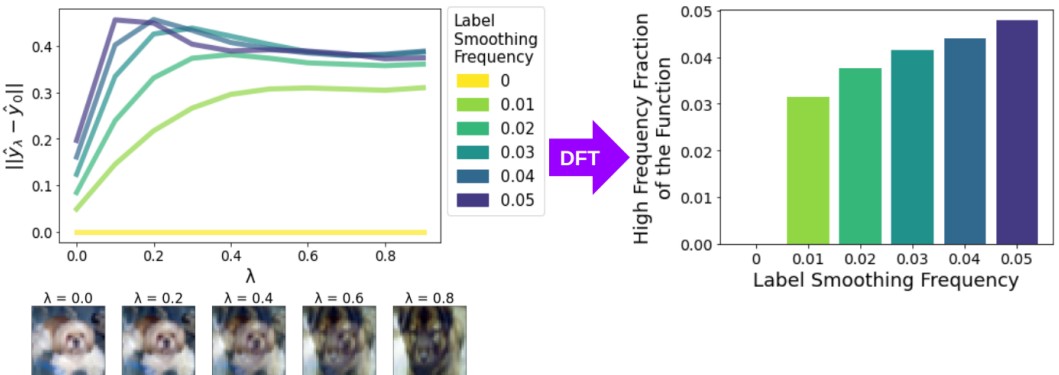

Figure 2: **Linear interpolation measurements on functions of varying frequency.** *Left*: Interpolation experiment between images of the same class on an oracle function: the true one-hot label perturbed by radial wave label smoothing of variable frequency. As the oracle function increases in frequency, the interpolating paths become less smooth. *Right*: Summary of this interpolation experiment via a discrete Fourier transform; as the frequency of the oracle function increases, so does the proportion of the DFT magnitude allocated to the high frequency components.

**Relationship between interpolation and label smoothing experiments.** In order to validate our interpolation methodology, we consider the oracle function that maps directly from the interpolated image $\mathbf{X}_\lambda$ to the target label used in our label smoothing experiments. By increasing the frequency of the label smoothing noise, we create oracle functions with higher frequencies. We then verify that our interpolation measurement indeed measures higher frequencies on the higher frequency oracle functions.

In Figure 2 *(Left)*, we show an example where we vary the label smoothing frequency from $f = 0$ to $f = 0.05$. We plot the value of $\lambda$ along the interpolating path on the x axis, and the norm of the difference between the model output on the interpolated image $\hat{y}_\lambda$ and the model output on the original image $\hat{y}_0$ on the y axis. The norm of the prediction difference allows us to visualize how smooth the function is along the interpolating path; smoother prediction difference norm indicates a lower frequency function.

In order to quantify the frequency of the function along the interpolating path, we compute the per-class discrete Fourier transform (DFT). Letting $\{\mathbf{X}_{\lambda_t}\}_{t=1}^T$ denote the set of interpolated images, the DFT coefficient for frequency $f$ is given by $\hat{Y}_f[m] = \sum_{t=1}^T \hat{y}_{\lambda_t}[m] \exp(-i2\pi f t)$, where $i$ is the imaginary number and $m \in \{1, \ldots, M\}$ denotes the class. In order to determine how the magnitudes of the DFT coefficients $\hat{Y}_f$ are distributed between low and high frequencies, we quantify how much of the spectrum is high frequency by choosing a frequency threshold and computing the fraction of the total coefficient magnitudes above the threshold. In Figure 2 *(Right)* we show this summary metric averaged over all within-class interpolating paths for the oracle functions.

Both sides of Figure 2 show that our interpolation methodology indeed measures the oracle functions that we constructed to be high frequency as high frequency. In Figure 2 *(Left)*, the higher frequency oracle functions have less smooth prediction norm differences, and in Figure 2 *(Right)*, they have a higher fraction of DFT coefficient magnitudes in the high frequencies.

### 3.3 DATA AND MODELS

Our experiments use the CIFAR-10 (Krizhevsky et al.) dataset of low-resolution natural images from ten classes, a mixture of animals and objects. Images are pixel-wise normalized by the mean and standard deviation of the training images. We consider six different convolutional neural networks that have achieved high accuracy on this task: two wide-resnets (Zagoruyko & Komodakis, 2017) of different sizes (wrn_32 and wrn_160), three shake-shake regularized networks (Gastaldi, 2017) of different sizes (shake_shake_32, shake_shake_96, and shake_shake_112), and pyramid_net, which uses the even stronger shake-drop regularization (Yamada et al., 2019). Our implementations are based

on Cubuk et al. (2019a); we use the same optimizer (stochastic gradient descent with momentum) and cosine learning rate schedule. We train without data augmentation (to ensure all models are trained on exactly the same examples), except for experiments that explicitly vary data augmentation. Without data augmentation, the test accuracies of our models are: 89.4% (wrn_32), 90.2% (wrn_160), 92.2% (shake_shake_32), 93.5% (shake_shake_96), 93.6% (shake_shake_112), and 95.8% (pyramid_net).

## 4 RESULTS

Throughout this section we show representative examples from our experiments to highlight our main findings. Full results (on all six models we tested) are included in section A.

### 4.1 SPECTRAL BIAS AND MODEL ARCHITECTURE

We begin by applying our label smoothing methodology to find that modern image classification CNNs exhibit spectral bias, learning low frequency target functions early in training and learning higher frequency functions as training proceeds. Note that Figure 3 *(Left)* shows spectral bias over a small but illustrative range of frequencies (the full range of frequencies we tested was 0 to 0.1); target functions of sufficiently low frequency are fit almost immediately and target functions of sufficiently high frequency are never learned during the 250 epochs of training we tested.

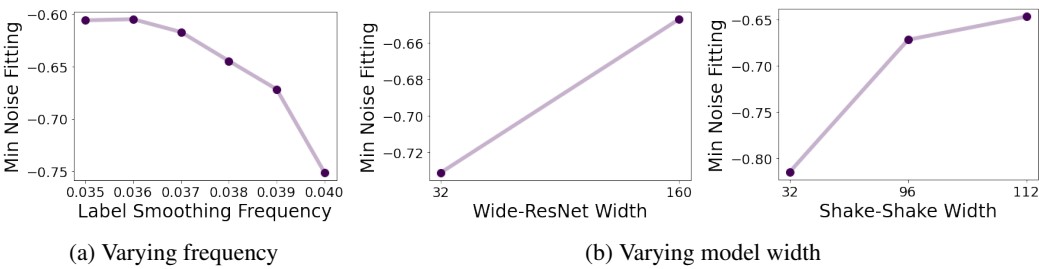

(a) Varying frequency        (b) Varying model width

Figure 3: **Modern CNNs exhibit spectral bias toward low frequencies, but larger networks learn high frequencies more readily.** Higher min noise fitting denotes that the label smoothing noise function is learned more readily. *Left*: Min noise fitting for a shake_shake_96 model with radial wave label smoothing of varying frequency; lower frequencies are learned more readily. *Center*: Min noise fitting for wide resnets of variable width at frequency 0.039; the larger model learns this frequency more readily. *Right*: Min noise fitting for shake-shake models of variable width at frequency 0.039; larger models learn this frequency more readily.

Although all models we tested exhibit spectral bias, we found that the precise nature of the bias depends on the choice of model. For example, with all else fixed, increasing the width of a model decreases its spectral bias, enabling it more readily fit target frequency functions. This trend is evident in Figure 3 for the wide-resnet family *(Center)* and the shake-shake family *(Right)*. Results of our label smoothing experiment on all six models we tested are included in section A.2.

### 4.2 SENSITIVITY TO NATURAL IMAGE DIRECTIONS

Although label smoothing with a radial wave offers a convenient global picture of spectral bias, by replacing the radial wave with other target functions we can use the same methodology to test more localized aspects of spectral bias. We take a step in this direction by considering the family of target label smoothing functions $S(\mathbf{X}; k) = \sin(2\pi f \langle \mathbf{X}, \mathbf{F}_k \rangle)$, where $\mathbf{F}_k$ is a diagonal Fourier basis image with frequency $k$ and the same dimensions as $\mathbf{X}$, visualized in Figure 4 *(Bottom)*.

We consider (a subset of) Fourier basis images because the Fourier spectra statistics of natural images are well studied (see, *e.g.* Tolhurst et al. (1992)): natural images tend to be composed of Fourier basis images with amplitude proportional to their inverse spatial frequency. Indeed, in Figure 4 we find that shake_shake_96 is more sensitive to label smoothing in low image frequency directions.

This finding is consistent with theoretical predictions of Basri et al. (2020) that models learn faster in regions of higher density of training examples: since low image frequencies are more common

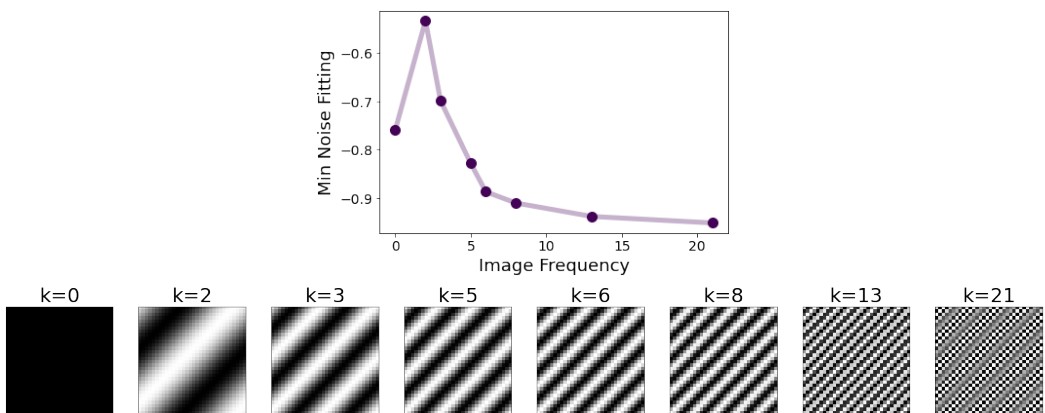

Figure 4: **Models are most sensitive to variations of low (but nonzero) image frequency, which are dominant in natural images.** *Top*: Effective noise fitting of a wrn_32 model with label smoothing of frequency 0.038 in various unit norm directions corresponding to Fourier basis images *(Bottom)* indexed by image frequency $k$ (scaled to [0, 1] for visualization).

in natural images, the effective sampling density enjoyed by a label smoothing target is higher in these directions. It is also possible that this bias is inherent to the convolutional model architecture regardless of the data distribution, as discussed in Ortiz-Jimenez et al. (2020b), or that a combination of both effects is at play. Determining the precise cause of this image frequency finding is an interesting direction for further study.

## 4.3 SPECTRAL BIAS AND REGULARIZATION

We also found a relationship between regularization and spectral bias, which is explored in Figure 5 and Figure 6. Some forms of regularization, like weight decay and dropout, are amenable to study by either of our methodologies; Figure 5 demonstrates agreement between these methods in the context of varying weight decay. Further results showing agreement between these two measurement methodologies are presented in section A.4.

In total we consider 4 types of (explicit and implicit) regularization: weight decay, training set size, data augmentation (Cubuk et al., 2019b;a; Zhang et al., 2018), and the strength of Mixup augmentation (Zhang et al., 2018). The latter three involve changes to the training data and are therefore only directly studied via our interpolation methodology, which we use in Figure 6.

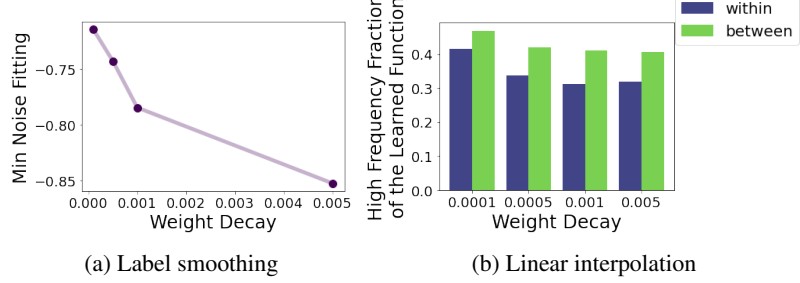

(a) Label smoothing  (b) Linear interpolation

Figure 5: **Weight decay inhibits learning high frequencies.** Results here are from shake_shake_32 using the radial wave label smoothing methodology at frequency 0.038 *(Left)* and linear interpolation within and between classes *(Right)*. Both measurement techniques show that increasing the penalty on the norm of the weights produces a stronger spectral bias, in the form of diminished fitting of high frequency noise and a lower-frequency learned function.

**Weight Decay.** Weight decay is a penalty on the square norm of the model parameters; a stronger penalty encourages a model to learn smaller-norm weights. Figure 5 shows that increasing the weight decay also encourages a model to learn high frequencies slower *(Left)* and to ultimately be lower-frequency *(Right)*. This result is consistent with the expectation that smaller-norm weights yield a learned function with a smaller Lipschitz constant and thus smoother changes in output (as discussed *e.g.* in Szegedy et al. (2014)). This experiment also demonstrates the nuanced relationship between the bandwidth or smoothness of the learned function and its accuracy; shake_shake_32 trained with weight decay 0.0005 or 0.001 achieves higher test accuracy compared to the same architecture trained with either more or less weight decay. Neither complexity nor smoothness is, in itself, purely beneficial or detrimental: the trick is to balance them appropriately for the dataset.

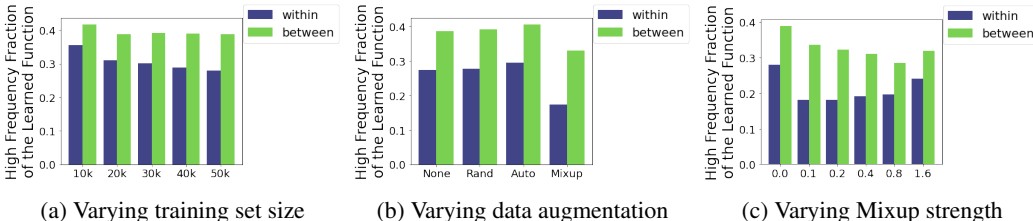

| (a) Varying training set size | (b) Varying data augmentation | (c) Varying Mixup strength |

Figure 6: **Implicit regularization in the form of dataset size and Mixup augmentation produces a lower-frequency learned function, but RandAugment and AutoAugment can increase function frequency.** *Left*: Interpolation experiment for wrn_160 trained with varying training set sizes: training with more examples produces a lower-frequency learned function, particularly within-class. *Center*: Training with Mixup (Zhang et al., 2018) makes wrn_160 lower-frequency, whereas training with RandAugment (Cubuk et al., 2019b) or AutoAugment (Cubuk et al., 2019a) makes it slightly higher-frequency. *Right*: Modestly increasing the Mixup strength produces lower-frequency learned functions, but further increasing the Mixup can reintroduce high frequencies within-class *(Top)*.

**Training Set Size.** As shown in Figure 6 *(Left)* on wrn_160, we find that increasing the number of training examples produces a lower-frequency learned function. Note that all models in Figure 6 *(Left)* were trained for the same number of epochs and with the same batch size, so the models trained with more examples were also trained for more gradient steps (since each epoch contained more batches). The models trained with more examples therefore had ample opportunity to fit the high frequencies present in their counterparts trained with fewer examples, and yet did not. This result supports the idea that additional examples serve as an implicit regularization by forcing the model away from more complex solutions that would predict the added examples incorrectly.

**Data Augmentation.** Data augmentation is a common strategy to increase the effective training set size without the expense of actually collecting additional examples. In Figure 6 *(Center)* we consider the effects of RandAugment (Cubuk et al., 2019b), AutoAugment (Cubuk et al., 2019a), and Mixup (Zhang et al., 2018) data augmentation strategies, each of which tends to improve the final accuracy of the trained model (with RandAugment producing the most substantial benefits in our experiments). RandAugment and AutoAugment generate new images by applying geometric and lighting transformations, whereas Mixup produces new images by linearly interpolating between pairs of existing images. We find that training with RandAugment or AutoAugment tends to make the learned function slightly less smooth between examples of the same class *(Top)*. Mixup (at strength 0.1) makes the learned function substantially smoother, both within-class *(Top)* and between-class *(Bottom)*. This experiment reinforces the nuanced relationship between function complexity and performance; a model can be improved by the addition of helpful complexity (as in RandAugment and AutoAugment, which encode robust transformation priors) or by the removal of unnecessary complexity (as in Mixup, which encodes a simple linear prior in unsupervised input regions).

**Mixup Strength.** Mixup augmentation (Zhang et al., 2018) perturbs each batch of training data as follows: each example is matched with a partner based on a random permutation of the batch, and each example is then perturbed towards its partner by an interpolation amount $\lambda$ drawn from a symmetric beta distribution (with the same $\lambda$ used for all examples in the batch). We refer to the parameter of this beta distribution as the Mixup strength, as it controls the degree to which the augmented images tend to lie close to an original training image or close to the average of two training images. A parameter

of 0 corresponds to no augmentation ($\lambda = 0$ or $\lambda = 1$, always using the original images), a parameter of 1 corresponds to the uniform distribution over $\lambda \in [0, 1]$, and a parameter of $\infty$ corresponds to $\lambda = 0.5$, the exact midpoint between a pair of training images. Figure 6 *(Center)* uses Mixup strength 0.1 (one of the values recommended by Zhang et al. (2018)). Figure 6 *(Right)* shows that, as expected, increasing the strength of the Mixup augmentation generally makes the learned function lower-frequency, but that Mixup that is too strong can yield high frequencies. Interestingly, the impact of Mixup strength on the learned frequencies is different within versus between classes, suggesting that perhaps tuning this parameter separately for these two cases may prove beneficial.

### 4.4 SELF-DISTILLATION

In a sense another form of regularization, we finally consider the effect of self-distillation on the smoothness of the learned function. In self-distillation, a 'teacher' model is trained with some form of strong regularization, in our case a combination of weight decay and early stopping based on training loss. A 'student' model with the same architecture is then trained from scratch to fit the pseudolabels produced by the teacher, instead of the original one-hot training labels (alternatively, an interpolation between the pseudolabels and the original labels may be used). Prior research (Furlanello et al., 2018) found that this procedure can train student models that outperform both their teachers and a baseline (the same architecture trained to completion on the one-hot labels).

Prior research has also sought to understand the mechanism behind knowledge distillation and self-distillation in particular. Mobahi et al. (2020) finds that self-distillation acts as a regularizer by limiting the basis functions available to the student to learn. We complement their theoretical work with our interpolation experimental methodology in Figure 7, where indeed we find that the student model learns a lower-frequency function than its teacher, when both have the same training loss. We conjecture that this effect is analogous to low-pass prefiltering common in digital signal processing: a high-frequency target function that we cannot adequately sample is first smoothed (in this case by being approximated by a regularized teacher) and then it can be modeled via samples without further loss in fidelity. Without this prefiltering, our samples are inadequate to capture the complexity of the target function, so we reconstruct an imperfect version corrupted by aliasing.

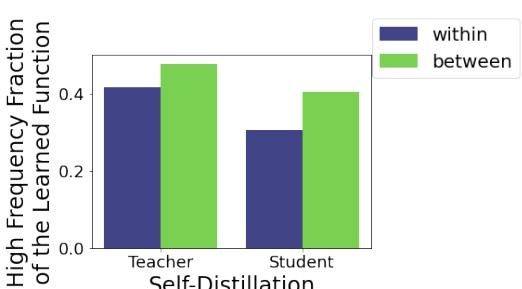

Figure 7: **Self-distillation produces a student model with a lower-frequency learned function than its teacher.** The teacher shake_shake_96 model is trained on one-hot labels and stopped early when training loss reaches a threshold. The student is trained to fit the pseudolabels produced by the teacher until the same training loss threshold is achieved, at which point it has higher validation accuracy than the teacher.

## 5 CONCLUSIONS

In this paper, we introduced two methods to measure spectral bias in modern image classification neural networks, and applied these methods towards the central question:

> *What kinds of function frequencies are needed for modern neural networks to generalize?*

Specifically, we applied these methods to examine the impact of a variety of training choices on the learned frequencies. Among training choices that improve validation accuracy, using a larger model, training longer, and using RandAugment (Cubuk et al., 2019b) or AutoAugment (Cubuk et al., 2019a) tend to increase the frequency of the learned function. However, regularizing with modest weight decay, training with more examples, using modest Mixup augmentation (Zhang et al., 2018), and performing self-distillation all improve validation accuracy while decreasing the frequency of the learned function. Our findings support the common wisdom that, while some complexity is necessary to fit the data, unnecessary complexity can harm generalization. Our experimental methods offer a valuable window into precisely what kind of function complexity we should strive for.

REPRODUCIBILITY STATEMENT

An anonymized copy of our code is included in the supplement. Our work uses the publicly-available CIFAR-10 dataset (Krizhevsky et al.).

ETHICS STATEMENT

Since our work is foundational rather than applied in nature, we do not anticipate any direct ethical concerns. However, as with any foundational research in machine learning, our work has the potential to improve the performance and controllability of machine learning models, particularly in the domain of image classification. These improvements in turn may find use in ethical (*e.g.* medical diagnostics, autonomous vehicles) or ethically questionable (*e.g.* surveillance) applications.

As for direct ethical impacts of our work, we acknowledge the energy costs of training and evaluating the models used in our experiments. One of our key contributions is proposing a method (via linear interpolation) to measure the frequency content of a learned function without retraining, which should enable future experimental research in spectral bias to be less energy-intensive.

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

# A APPENDIX

## A.1 LINEAR INTERPOLATION: METHODOLOGICAL DETAILS

For each sampled path, we compute the discrete Fourier transform (DFT) of the prediction function separately for each of the $M$ class predictions, take the (real) magnitude of the resulting (complex) DFT coefficients, and average them among the $M$ classes. We then average these coefficient magnitudes among the many paths (all within-class paths and, separately, all between-class paths) and compute the fraction of this averaged DFT magnitude that is allocated to the high frequency components. We use a simple threshold that considers the lowest 10% of frequency components (*i.e.* frequencies between 0 and 0.05) "low" and the remaining (between 0.05 and 0.5) "high", but qualitatively our results are not sensitive to this particular threshold. The highest frequency we can measure is 0.5, the Nyquist rate corresponding to our sampling distance of 1; if a particular maximum frequency is desired for measurement all that must be changed to achieve it is the sampling distance along the interpolating paths.

## A.2 LABEL SMOOTHING

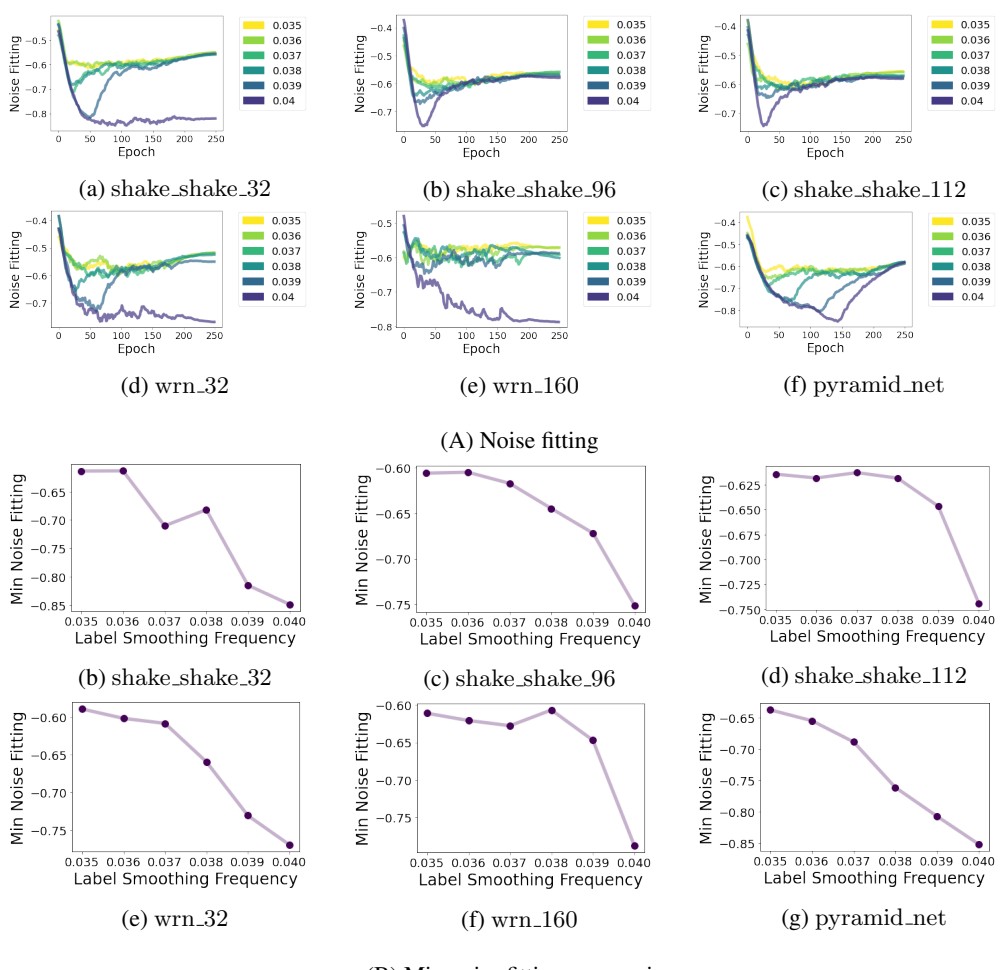

(A) Noise fitting

(B) Min noise fitting summaries

Figure 8: **All six models we tested exhibit spectral bias.** Here we show noise fitting when training each model with different frequencies of radial wave label smoothing.

## A.3 SENSITIVITY TO NATURAL IMAGE DIRECTIONS

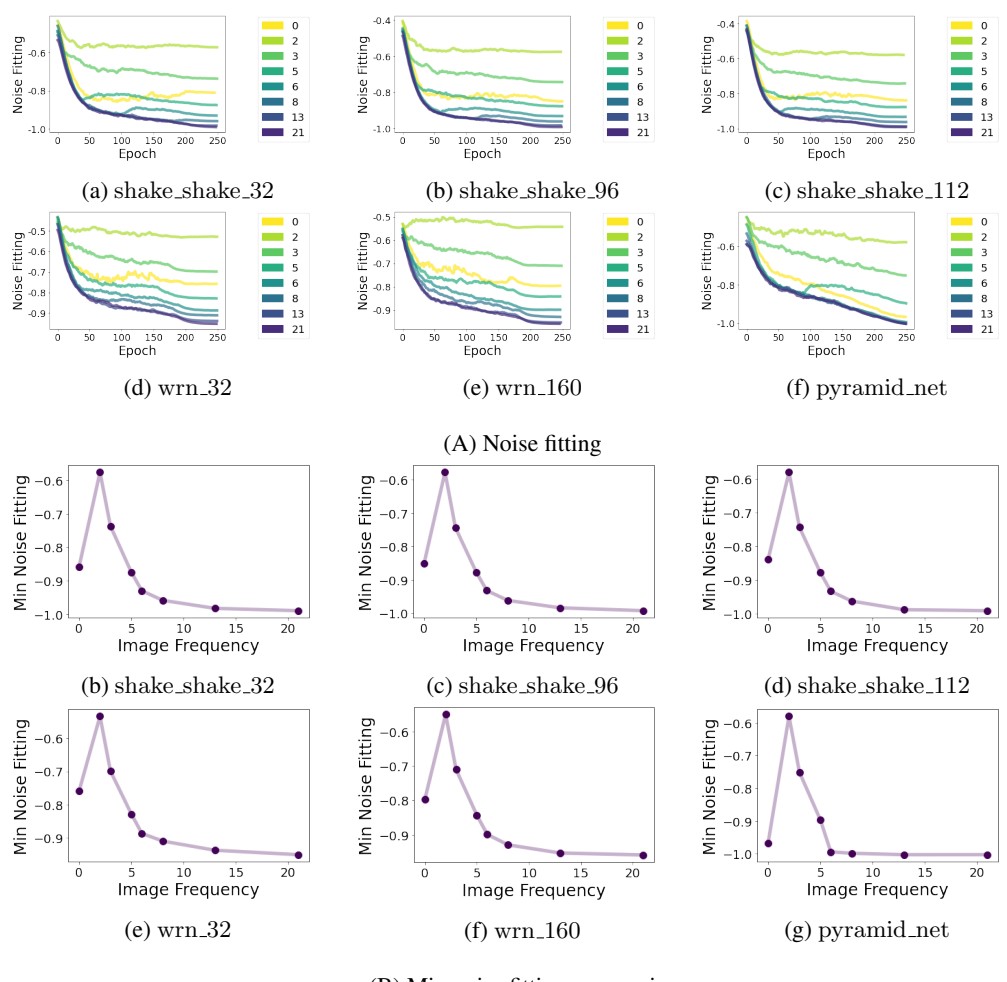

(a) shake_shake_32    (b) shake_shake_96    (c) shake_shake_112

(d) wrn_32    (e) wrn_160    (f) pyramid_net

(A) Noise fitting

(b) shake_shake_32    (c) shake_shake_96    (d) shake_shake_112

(e) wrn_32    (f) wrn_160    (g) pyramid_net

(B) Min noise fitting summaries

Figure 9: **All six models we tested exhibit sensitivity to variations of low (but nonzero) image frequency, which are dominant in natural images.** Here we show noise fitting when training each model with label smoothing of frequency 0.038 in various unit norm direction corresponding to Fourier basis images indexed by frequency $k$.

## A.4 AGREEMENT BETWEEN LABEL SMOOTHING AND LINEAR INTERPOLATION

### A.4.1 VARYING THE FREQUENCY OF THE RADIAL WAVE LABEL SMOOTHING

When the frequency of the radial wave used for label smoothing increases, models take more time to fit the smoothing noise. Figure 3 *(Left)* and Figure 8 show this using label smoothing; the corresponding interpolation experiment is shown in Figure 10. The target frequencies between 0.035 and 0.04 are close enough that, were the model to fit each perfectly, the interpolation curves would be nearly visually indistinguishable, as we can tell from Figure 2 *(Left)*. However, the model is actually smoother when trying to fit higher frequency label smoothing noise, because it fits this noise less well (in addition to fitting it later in training). We can see this effect, for instance, by noting that wrn_32, wrn_160, and shake_shake_32 fail to fit frequency 0.04, both in Figure 8 and Figure 10.

It is also worth noting that, in Figure 10 and repeatedly across our interpolation experiments, the learned function is smoother (lower-frequency) within-class and less smooth (higher-frequency) between-class. This is to be expected since the model must change predictions somewhere along the path between examples from different classes.

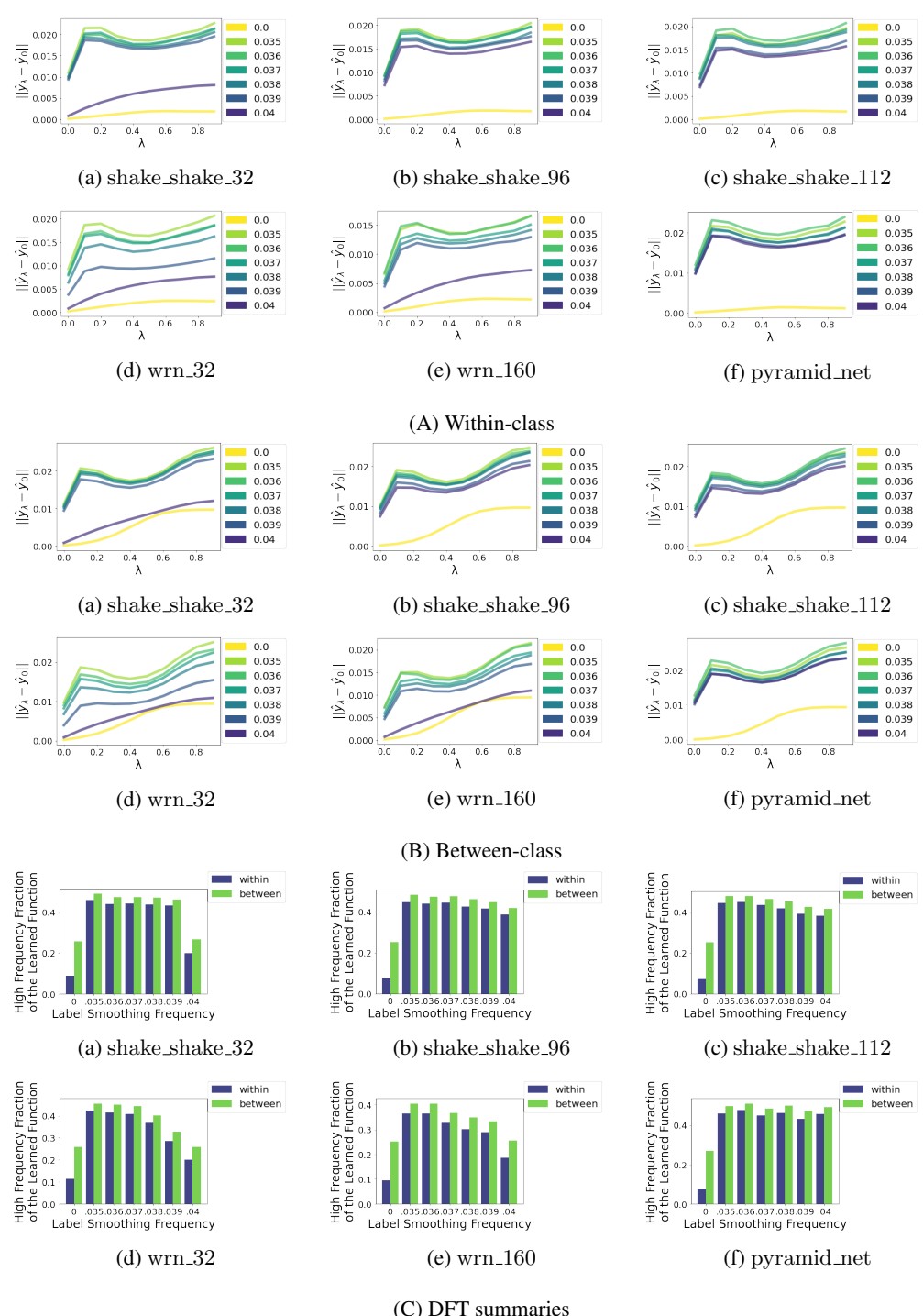

Figure 10: **Spectral bias is evident via interpolation when training with radial wave label smoothing at various frequencies.** Results here parallel those in Figure 8 but using the linear interpolation methodology.

### A.4.2 LEARNING LOW FREQUENCIES FIRST

The label smoothing experiment presented in Figure 3 *(Left)* and Figure 8 shows that low frequency target functions are learned earlier than higher frequency targets; we can also confirm this finding

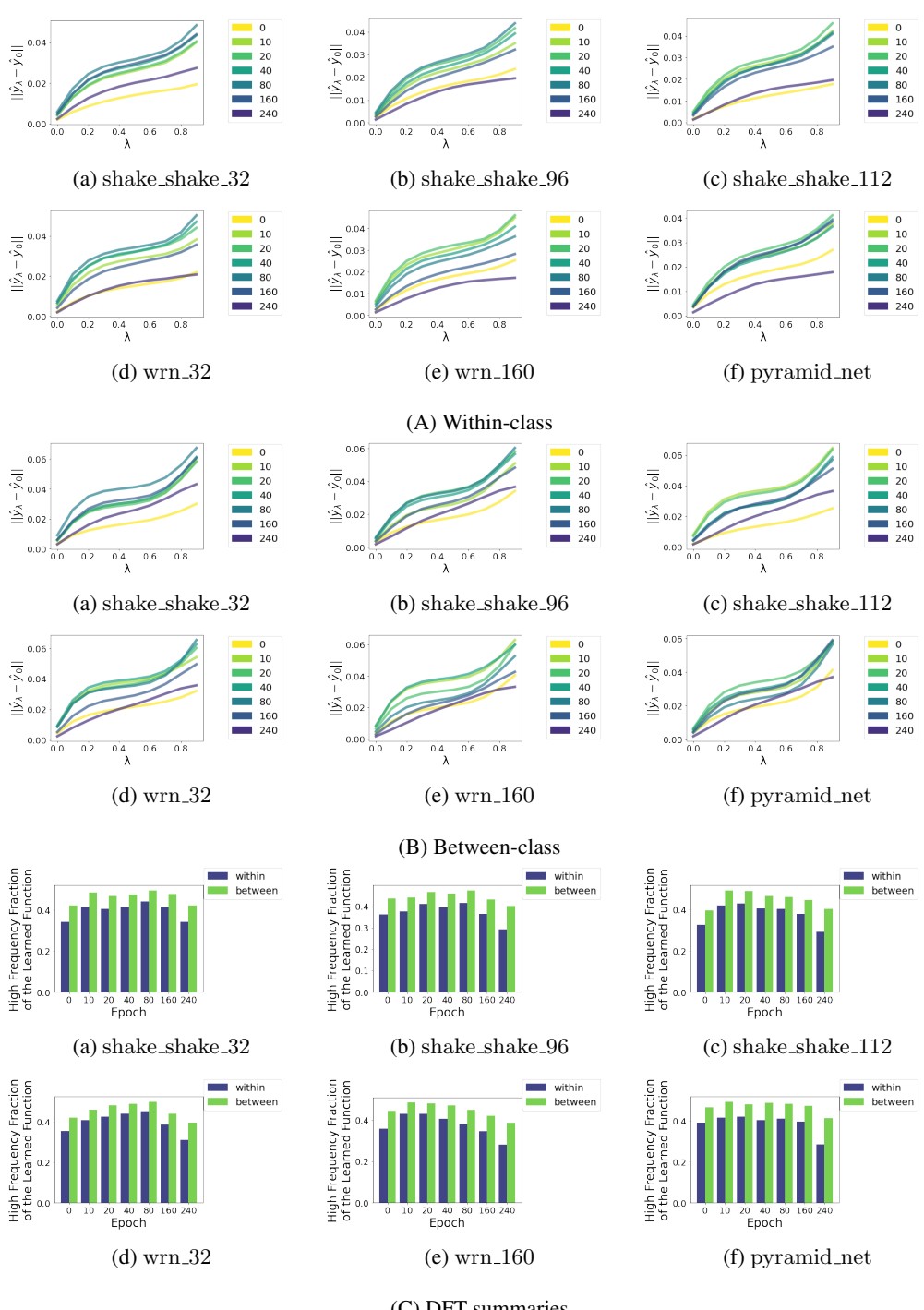

Figure 11: **Spectral bias is evident via interpolation when training as usual and comparing checkpoints at different epochs.**

using interpolation with model checkpoints saved at different epochs throughout training (without any label smoothing). We see that indeed in the early stages of training low frequencies predominate, with higher frequencies entering as training proceeds. Surprisingly, we also see that towards the end of training low frequencies again predominate; the reason for this is an interesting question for further research.

### A.4.3 WEIGHT DECAY

Figure 5 shows on shake_shake_32 via both label noise and linear interpolation that increasing the weight decay results in a smoother (lower frequency) learned function. Figure 12 shows the same result across all six models we tested.

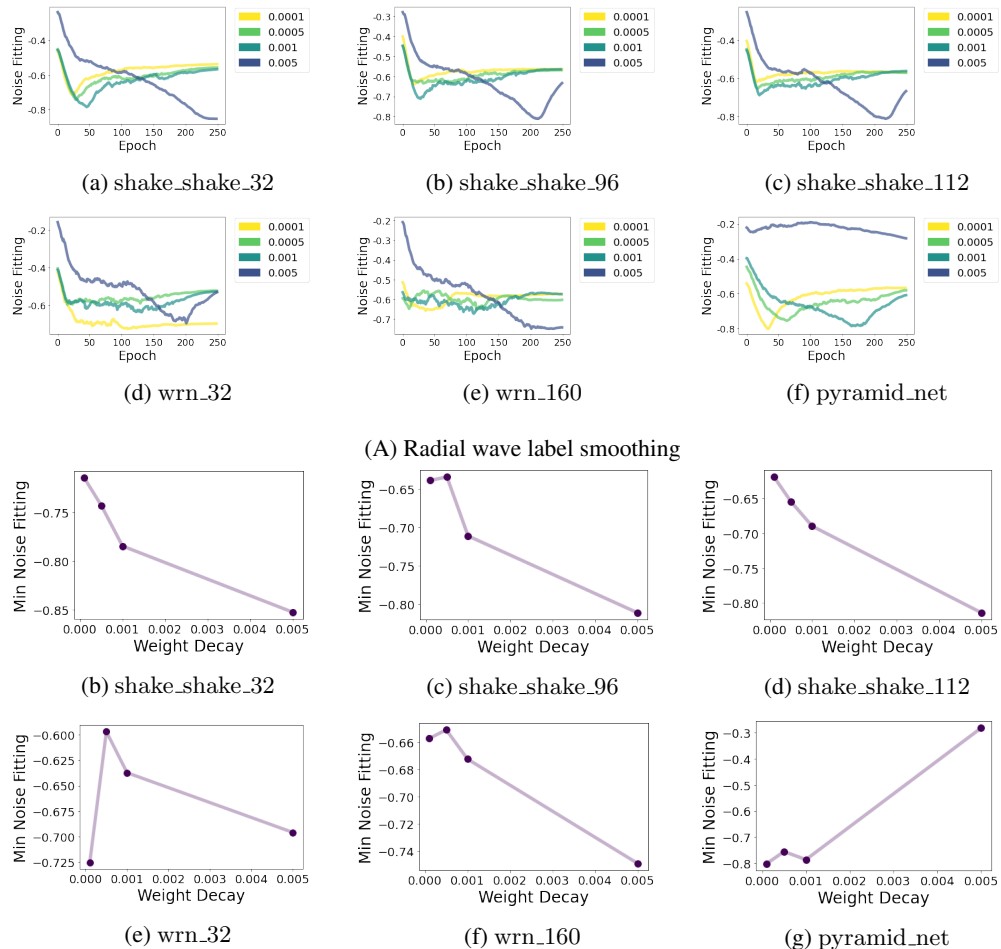

(A) Radial wave label smoothing

(B) Min noise fitting summaries of label smoothing. Note that for pyramid_net, min noise fitting does not fully capture the results of noise fitting; the "dips" are clearly separated across epochs, with higher weight decay inducing delayed dips, even if they achieve similar minimum noise fitting values.

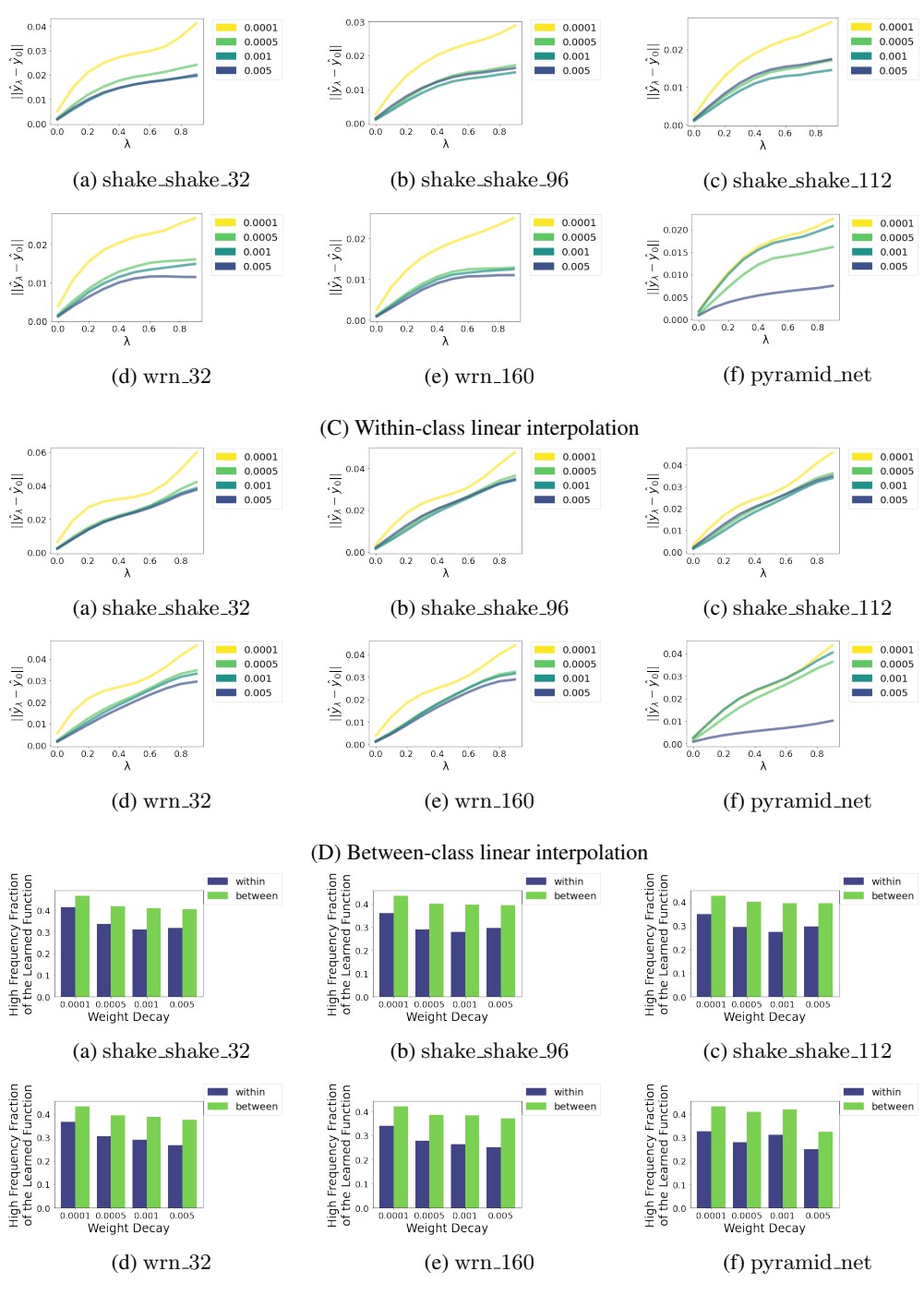

(C) Within-class linear interpolation

(D) Between-class linear interpolation

(E) DFT summaries of linear interpolation

Figure 12: **Weight decay increases spectral bias, producing a lower-frequency learned function.**

## A.5 TRAINING SET SIZE

Figure 6 *(Left)* shows the regularization (frequency reduction) effect of increasing dataset size for wrn_160. Figure 13 shows the same effect on all six models we tested.

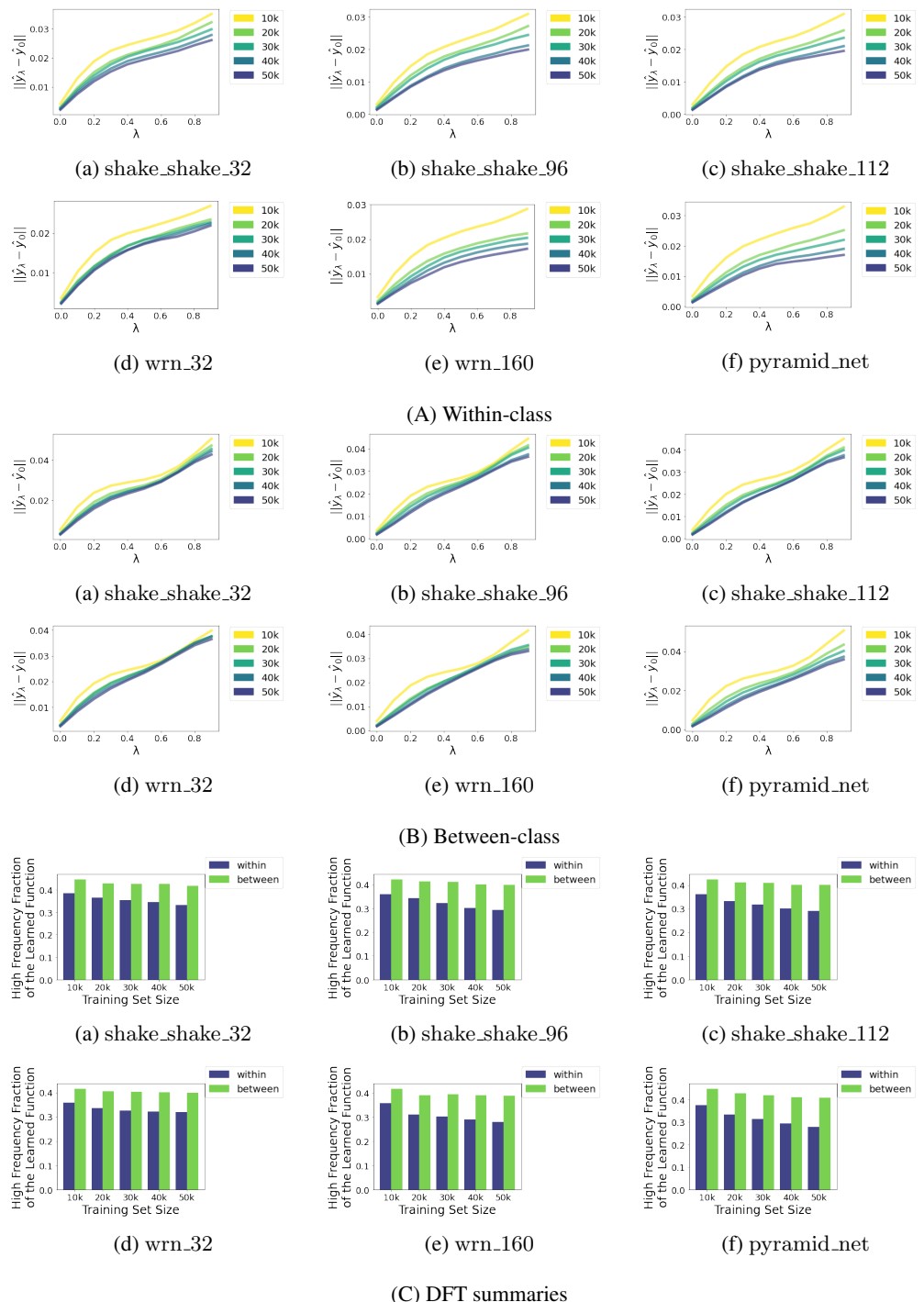

Figure 13: **Increasing the number of training examples reduces the frequency of the learned function.**

## A.6 DATA AUGMENTATION

Figure 6 *(Center)* uses linear interpolation to study the effect of common data augmentation procedures on the learned frequencies for wrn_160. Figure 14 shows the same experiment on all six models we tested.

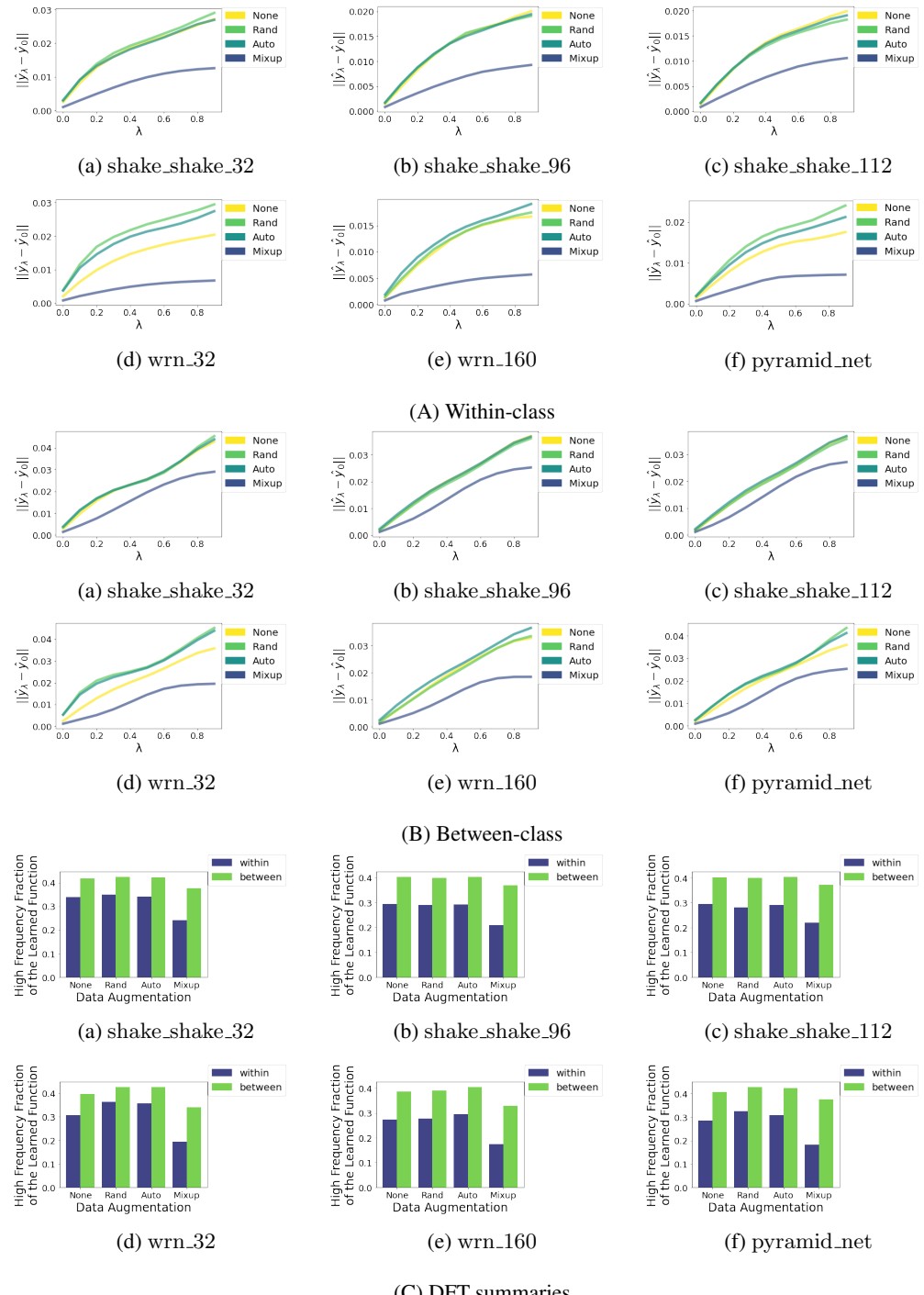

Figure 14: **Mixup augmentation produces a lower-frequency learned function on all models tested. On some models, RandAugment and AutoAugment produce a higher-frequency learned function.**

### A.6.1 Mixup Strength

Figure 6 *(Right)* uses linear interpolation to show that training with Mixup data augmentation (Zhang et al., 2018) causes wrn_160 to learn a lower-frequency function, but too much Mixup can produce a higher-frequency function. Figure 15 repeats the experiment on all six models we tested.

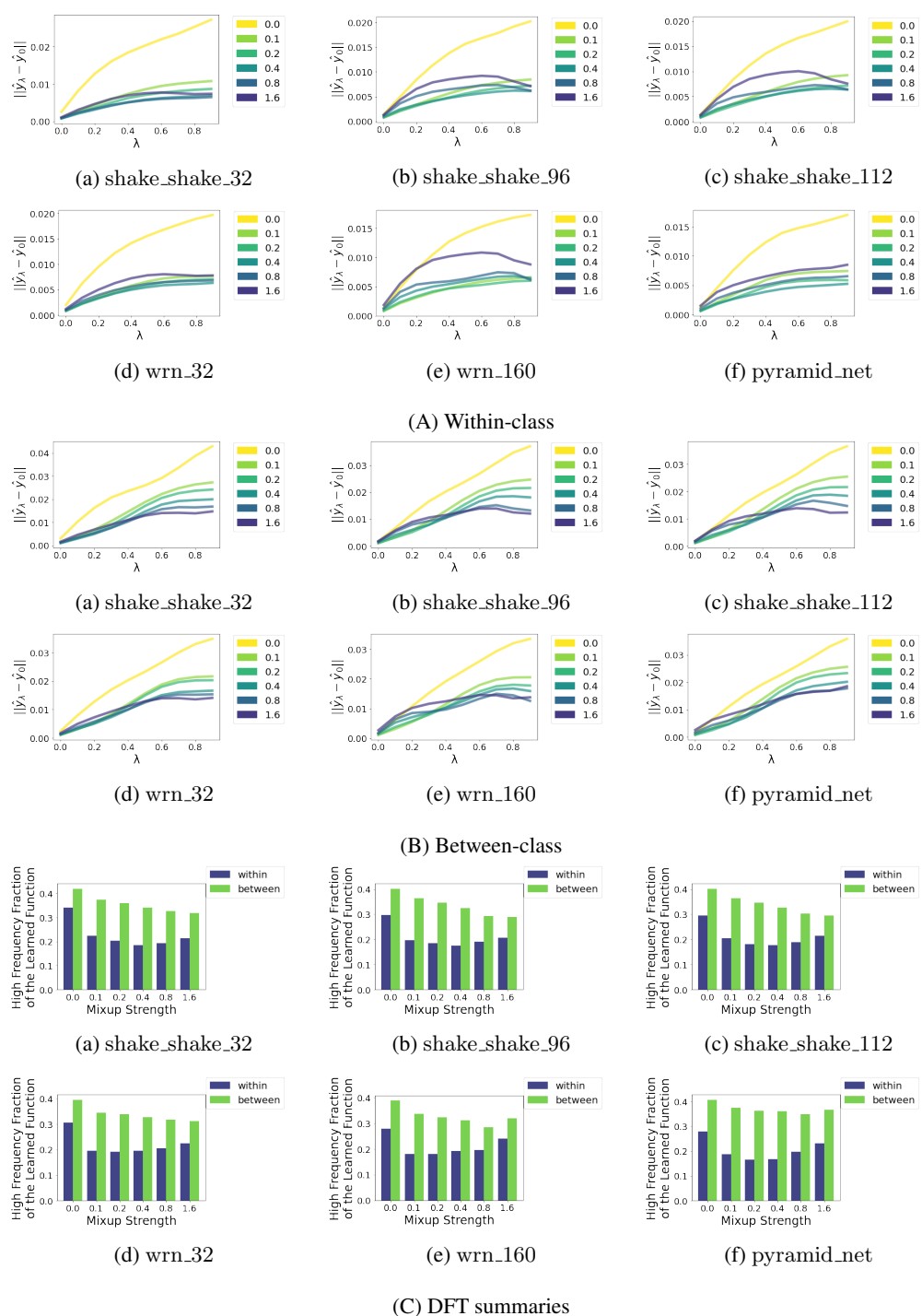

Figure 15: **Modest Mixup augmentation produces a lower-frequency learned function, but Mixup that is too strong can induce higher frequencies.**

## A.7 SELF-DISTILLATION

Figure 7 uses linear interpolation to show that self-distillation with shake_shake_96 produces a student model that is lower-frequency than its teacher, even though both are trained to the same training loss and the student has higher validation accuracy than the teacher. Figure 16 shows the same result across all six models we tested.

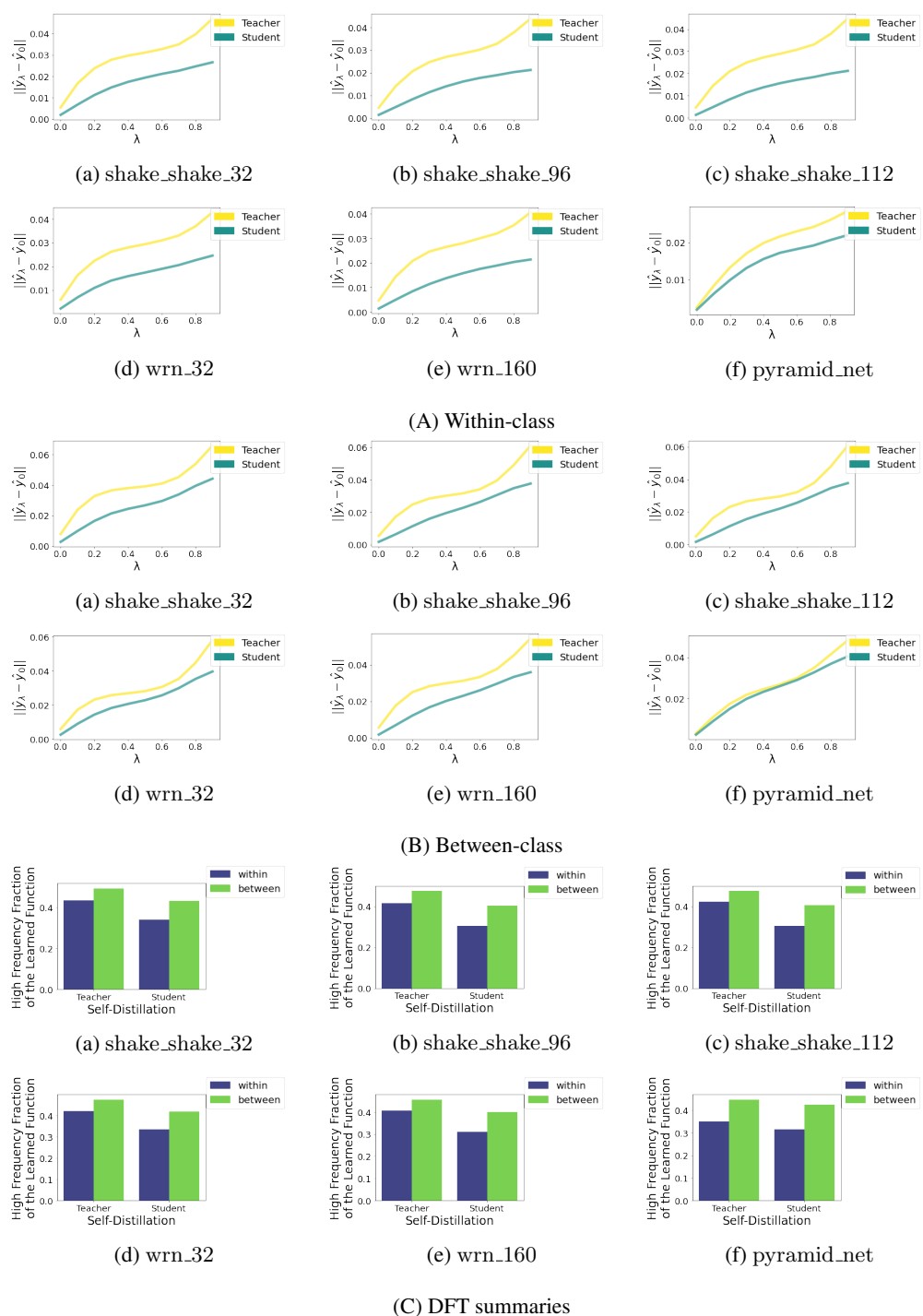

Figure 16: **Self-distillation produces a lower-frequency student compared to its teacher.**

