# OpenReview forum: "Spectral Bias in Practice: the Role of Function Frequency in Generalization"
_ICLR.cc/2022/Conference — ICLR 2022 Submitted_

### Official Review · Reviewer_axgN · 2021-10-26

**Correctness:** 3
**Technical Novelty And Significance:** 1
**Empirical Novelty And Significance:** 2
**Recommendation:** 3
**Confidence:** 4

**Main Review:**

# Strengths:

-  The text is mostly readable (apart from a few parts as denoted below).
-  The links to model distillation are interesting, but not explored enough.
-  The motivation is clear.

# Clarification questions:

-  How is sec. 3.1 (fig. 1) different to classic overfitting?
-  In the first paragraph of sec. 3.2 the paper mentions ‘grouping of images in pairs’, however it is not mentioned in that paragraph how the grouping is done.
-  What is the ‘effective amount of interpolation’ mentioned in sec. 3.2?
-  One of the most popular networks (and the almost default baseline) is ResNet, so it is surprising that ResNet is not used. In addition, what are the clean accuracy of the networks used?
-  Why are ablation studies on different regularization schemes (sec. 4.3) conducted in different networks? Why are wide-residual nets preferred for weight decay and shake-shake for the training set size?
-  Why does RandAugment add ‘necessary complexity’? How does RandAugment differ from Mixup or other training schemes that demonstrate low-frequency preference?
-  Frequently, when training image recognition networks on CIFAR10 data augmentation is used, however it is not explicitly mentioned (other than in sec. 4.3). Was there any data augmentation used in sec. 4.1 and 4.2?
-  In sec. 4.2, is there some intuition for stopping at k=21 (fig.4)?


# Improvements:

-  The paper includes empirical results and decisions without mentioning any motivation or theoretical grounding for those. For instance, see the comment about different networks in the ablation studies, or why those networks were used over state-of-the-art networks or the popular ResNet. In addition, neither an algorithm or the source code is provided, which makes it harder to assess the decision choices on the code and the plots.
-  I would recommend sharing further information on the function S (sec. 3.1); sinusoidal might not be ideal for image domain applications, which is again not mentioned or discussed in the paper.
-  The paper focuses on a single experiment, i.e. image classification on CIFAR10, which makes it hard to appreciate whether it generalizes beyond this setting. For instance, do the same results emerge in self-supervised learning?
-  In a similar vein to the previous comment, it would be more convincing if the results on different datasets (e.g. STL10, Imagenet) were presented along with additional training algorithms to demonstrate that the observations indeed generalize well.
-  One implicit assumption of the paper is that CIFAR10 shares the same high-frequency statistics as the datasets used in practice. Do higher-dimensional datasets (e.g. Imagenet, CelebA-HQ, COCO, iNaturalist) share the same statistics and do the properties presented here generalize to those?
-  The sec. 3.2 (which is probably the most critical in the method) could be improved in terms of writing. Currently, a number of questions are not answered. For instance: why do the path lengths vary? What do the different colors in Fig. 2 depict? Is y_lambda (fig. 2) and y_t (sec. 3.2) different?
-  Is there some theoretical foundation on how the rapid changes in the cumulative difference of the norm (sec. 3.2) are related to the high-frequency components?


# Minor comments:

-  The ‘shake-shake’ (sec. 3.1) is mentioned without defining that it is a network or providing any citation; the reader needs to go all the way to the experiments to understand what this refers to.
-  Lambda is not a formula (sec. 3.2).
-  The legends are currently broken, e.g. in Fig. 3 they are blocking a critical part of the figure (i.e. > 200 epochs).
-  Is the lambda mentioned in sec. 4.3 (Mixup strength) the same as in sec. 3.2?


**Summary Of The Paper:**

The paper focuses on the spectral bias of neural networks, i.e., their tendency to learn first the low-frequency information. The goal of this work is to extend the spectral bias into practical image recognition networks, i.e., beyond the fully-connected nets and the NTK regime that it was previously studied. To that end, the paper conducts a number of studies in both explicit and implicit regularization schemes used in practice, and provides links for the success of model distillation.

**Summary Of The Review:**

In my opinion, the spectral bias can be a useful tool in analyzing the inductive bias of networks, but the current format of the manuscript is not ready for acceptance. There are a number of questions on the methodology and how this relates to the spectral bias, that is the main contribution of this work. In addition, the experiments are not considering several cases (e.g. other datasets, or alternative learning algorithms) that might have correlations with their results. Furthermore, previous results about the manifold of the data (ie. that it impacts the frequencies learnt) are not discussed in the paper. If the revised version addresses the improvement points, I can reconsider my rating.

---

> ### Author Response · Authors · 2021-11-12
> **Response to reviewer axgN (part 1 of 2)**
>
> We thank the reviewer for their time and feedback on our paper. Multiple reviewers suggested editing the presentation of our results to be quantitative rather than qualitative, and easier to interpret. Accordingly, in a separate comment for all reviewers, we provide aggregated metrics that summarize our findings quantitatively in additional figures. We will update the figures and text of the main paper as well to focus on clarity of presentation. We respond to reviewer-specific comments below.
> - Difference vs. classic overfitting: Note that in Fig. 1 around epoch 150, the noisy validation loss drops, in tandem with the (noisy) training loss. The noisy validation loss is computed on the regular validation images, with labels perturbed by the same label smoothing function as the training labels. Because the noisy validation loss tracks the noisy training loss, we know that the model is fitting our target noise function, and not just overfitting to the training data (as in classic overfitting).
> - Grouping images in pairs: We consider two types of pairings, within-class and between-class. For within-class pairings, for each class we choose 200 pairs of images from the validation set, at random and without replacement. For between-class pairings, for each of the 45 possible pairs of classes (10 choose 2) we choose 200 pairs of images at random, where one image belongs to each class in the pair. We also include an anonymized copy of our code if the reviewer would like to see our implementation of this procedure.
> - Effective amount of interpolation: This refers to the value of lambda, the fraction of the total interpolation distance that describes each interpolated image. For example, the test images being interpolated have lambda values of 0 and 1, respectively, and an image halfway between them has lambda=0.5. We illustrate lambda in Fig. 2 (bottom).
> - Choice of models: We used the same architectures (including wide-resnet) as in the AutoAugment paper [Cubuk et al., 2019a], which enables easily studying the effects of data augmentation. Our methodology should extend easily to any architecture; we plan to include results on additional models in the final version of the paper. The clean test accuracies of our models (without data augmentation) are: 89.4% (wrn_32), 90.2% (wrn_160), 92.2% (shake_shake_32), 93.5% (shake_shake_96), 93.6% (shake_shake_112), and 95.8% (pyramid_net).
> - Different networks in different figures: We presented results on different models in the main text just to give a sense of the breadth of models we tested, although the claimed results hold across all the models we tested. Full results of each ablation study on each model are included in the appendix.
> - Specifics of data augmentation: We will add a more substantial description of each of the data augmentation methods we study to the main text, as this is important background to understand our statement about “necessary complexity”. AutoAugment and RandAugment produce augmented images by applying various perturbations (like translations, rotations, and lighting changes) to the original training images, adding the “necessary complexity” that models should be robust to these perturbations we know are irrelevant for the task of image classification. In contrast, Mixup augmentation produces augmented images by linearly interpolating both the images and the labels of original training examples, introducing the prior that the model’s predictions should vary smoothly.
> - Data augmentation in Sec. 4.1 and 4.2: We did not use data augmentation for these experiments. The reason for this is that the label smoothing methodology requires modifying the training labels, so for fair comparison we want to ensure that all the models we compare with this method are trained on identical training images.
> - Maximum image frequency: Yes, the reason to stop at k=21 is that CIFAR10 images are low resolution, so the maximum image frequency they can include is inherently limited. The theoretical maximum image frequency in this diagonal direction is actually just over k=22 (because the resolution of the image is 32x32, so there are 16 positive frequencies in each dimension and the norm of the vector [16, 16] is just over 22). This is illustrated in Fig. 4 (bottom); increasing the image frequency further would result in variations that are smaller than the scale of a pixel.

---

> > ### Author Response · Authors · 2021-11-12
> > **Response to reviewer axgN (part 2 of 2)**
> >
> > - Design choices and code: We include an anonymized copy of our code for the reviewers to see our exact implementation of each experiment. We hope the reviewer’s other questions about our design choices have been answered above; if not, please follow up and we are happy to provide further details.
> > - The function S: Perhaps we do not fully understand the reviewer’s concern (feel free to follow up if this is the case). We use a sinusoidal function specifically because it is the canonical periodic function (with a well-defined frequency) and we wish to study the bias of neural networks with respect to function frequency. Our choice of S is also similar to that used in [Rahaman et al., 2019], with the extension that we apply S to label smoothing for multiclass classification, and consider sinusoids in different directions (to study the interplay of function frequency and image frequency).
> > - Generalization to other settings: We agree this is an interesting direction for future work. There are many potential regimes in which we hope to study spectral bias in the future; the primary purpose of this paper is to introduce general-purpose tools for studying spectral bias in practice, and to explore some ways in which measuring this bias can shed light on model behavior.
> > - Different datasets and algorithms: We agree this is an important direction for future work. We plan to apply our methods to a broader range of image classification datasets and model architectures.
> > - Image statistics: The image frequency statistics of natural images are well studied and remarkably robust across datasets; see e.g. [Tolhurst et al., 1992]. The only limitation (to the best of our knowledge) of low-resolution images like those in CIFAR10 is that their maximum image frequency is limited by resolution (as we discussed above). However, natural images have low power in the high frequencies, so most of the spectrum is preserved even at low resolution.
> > - Clarity of Sec. 3.2: We will endeavor to clarify the writing of this section, as it is indeed important for understanding our subsequent linear interpolation results. Path lengths vary because we take samples at regular intervals (in terms of image norm), and different pairs of images have different image norm distances between them; if the images are farther apart then we will use more samples in our interpolation. We will clarify the legend of Fig. 2 (it is described in the caption); the different curves represent the target function defined by label smoothing with different frequencies (the number in the legend is the frequency of the label smoothing). Y_lambda and y_t both denote the model predictions at an interpolated image, but using different parameterizations: t denotes the absolute interpolation distance (the norm of the difference between this image and the first image in the path) whereas lambda denotes the relative interpolation distance (as a fraction of the total path length).
> > - Interpreting interpolation results as frequency: This is an important point that we will emphasize in the main text. A low frequency function is by definition slowly-varying; a high frequency function is by definition rapidly-varying. This fact is visualized in two dimensions in the Fourier basis images shown in Fig. 4 (bottom): the low frequency functions (on the left) vary much more slowly than the high frequency functions (on the right).
> > - Using ‘shake-shake’ before defining it: Thanks for pointing this out; we will define the network architectures earlier in our revision.
> > - Lambda as a formula: Perhaps the reviewer misread “lambda *in* the interpolation formula” as “lambda *is* the interpolation formula”? Indeed lambda is a variable, not a formula (the current wording is correct, although we will endeavor to make it more clear).
> > - Legends overlapping figures: Thanks for pointing this out. We note that the portions of the figures that are essential for understanding the results (the relative ordering of the curves) are never occluded; nevertheless we will revise all the figures to ensure legends do not occlude figures.
> > - Lambda in Sec. 4.3 and Sec. 3.2: Yes, in both sections lambda refers to the interpolation fraction (between 0 and 1).
> >
> > Nasim  Rahaman,  Aristide  Baratin,  Devansh  Arpit,  Felix  Draxler,  Min  Lin,  Fred  A.  Hamprecht, Yoshua Bengio, and Aaron Courville. On the spectral bias of neural networks, 2019.
> >
> > DJ Tolhurst, Y Tadmor, and Tang Chao.  Amplitude spectra of natural images. Ophthalmic and
> > Physiological Optics, 12(2):229–232, 1992.

---

> > ### Comment · Reviewer_axgN · 2021-11-26
> > **Response to the rebuttal**
> >
> > I am thankful to the reviewer for their responses. I have studied carefully the updated manuscript and the responses to the reviewers, and I have the following remarks:
> >
> > I still *do not find a clear motivation for the sinusoid*. The responses mention that it is chosen since it has a well-defined frequency, it is periodic. Many other functions have a well-defined frequency, as for periodicity, it seems this is not used in the current manuscript (please mention if this is not the case). Another reason is that the previous work in spectral bias is using this function: yes, but for the functions there which are synthetic-like, it does make sense from a signal processing perspective. I am still not sure what the motivation in images to use this particular transform are though.
> >
> > In addition, as the original review mentioned, I am not sure *which of the observations can be validated outside of CIFAR10*. This was explicitly mentioned in the original review, but the responses also validate this point. For instance, the low-resolution images constrain the maximum frequency that can be used in the experiments (k=21 answer). In addition, a strong point for using other datasets, is that CIFAR10 has a) only a single central object in each image and b) it has very discrete classes. If one of the contributions of this work is to demonstrate 'spectral bias in practice', I am not sure whether the outcomes of CIFAR10 can be validated in other datasets, e.g. STL10, CIFAR100, Tiny-Imagenet etc.
> >
> > In addition, one of the main claims of the existing paper is *the study of different regularization and training schemes* and how they perform with the established metrics [also, mentioned in the response to the reviewer tm3W]. I do not believe this qualifies as a high enough bar for ICLR, since those might be correlated with the dataset they have been tried on (see comment above on CIFAR10).
> >
> > Also, I find that the paper has few issues with *scattered notation* even after the revision. For instance, in sec. 3.1 the $\bf X$ is defined as a tensor, then the norm $\|\| \bf{X} \|\|$ is defined (sec. 3.2); is this a tensor norm or was it supposed to be on the vectorized $\bf X$? Despite the concerns on Fig. 1, it still has different scales in the y axis, which makes it hard to compare the curves explicitly. In sec. 3.2 the $\hat{y}_{\lambda}$ should be bold, right? The $i$ is used as an index in sec. 3.1 and then as the imaginary number in sec. 3.2.
> >
> > Also, three clarification questions:
> >
> > 1. In sec. 3.2 I believe each of the plotted curves in Fig. 2 is a separately trained network with ground-truth labels defined as in sec. 3.1 right?
> > 2. The experiment in sec. 3.2 seems to account for the complexity only in the output space, right?
> > 3. Why is the f=0.005 considered 'high' frequency in this work?

---

> > > ### Author Response · Authors · 2021-11-28
> > > **Response to the response (part 1 of 2)**
> > >
> > > We thank the reviewer for their interest in our work and continued engagement with our revised manuscript. We respond to questions/comments below; please follow up if anything is still unclear.
> > >
> > > Motivation for the sinusoid: We are interested in studying the frequency decomposition of the neural network function, a function from image to class probabilities. The canonical frequency decomposition of any (square integrable) function is the Fourier transform, which decomposes the function into a sum of sinusoids at different frequencies. Indeed, sines and cosines form the canonical orthogonal basis functions based on frequency, and are therefore a very natural choice. We also wish to clarify that the sine wave used in Rahaman et al., which is the inspiration for our choice of label noise function, *is* used in the context of image classification (see Experiment 3 in Rahaman et al., which uses a binary subset of MNIST and sinusoidal label noise). The key difference between their label noise experiment and our label smoothing experiment is not in the domain (both consider images), but in the number of classes (Rahaman et al. consider only binary classification with MSE loss, whereas we consider multiclass classification with cross-entropy loss). For both of these reasons (it is the canonical single-frequency basis function, and the choice used in prior work studying spectral bias in image classification), we believe that a sine wave is the natural choice to study spectral bias in our label smoothing experiments. If the reviewer has another function in mind that they believe is better-suited to studying spectral bias for image classification, please let us know what it is and why it might be more suitable.
> > >
> > > Other datasets: We agree that studying spectral bias in a wide range of domains, including different datasets (e.g. higher-resolution images, natural language, audio), model architectures (e.g. transformers, implicit neural networks), and types of training (e.g. contrastive learning, self-supervised learning) is an important next step, and we hope that the methods we propose in this paper will facilitate this further exploration. However, we believe that this broad application of our methods to further contexts is beyond the scope of the current conference paper and better done thoroughly in followup work. Nonetheless, we believe that the paper with its current scope makes a valuable contribution both by introducing novel and practical methods to measure spectral bias and by applying these methods to a wide variety of models and training schemes on CIFAR10.
> > >
> > > The reviewer also mentions a concern that “different regularization and training schemes and how they perform with the established metrics…might be correlated with the dataset they have been tried on.” It is certainly true that model architectures and training schemes are commonly adapted to specific datasets, but this adaptation has no direct bearing on our experimental results. For each type of regularization and training method we consider, we explicitly sweep over the relevant hyperparameter and then compare the effect of each hyperparameter value on the measured function frequencies. For instance, even if a certain value of weight decay is optimized for use on CIFAR10, we test a range of weight decay values and study how changing the weight decay affects the frequency of the learned function. Please note also that we do not claim results beyond the experiments we present; although we expect many of our findings to generalize to other image datasets, this is beyond the scope of our claims in the current paper.

---

> > > > ### Author Response · Authors · 2021-11-28
> > > > **Response to the response (part 2 of 2)**
> > > >
> > > > Scattered notation:
> > > > - By $||\textbf{X}||$ we refer to the 2-norm of the vectorized image. This clarification has been added to the text (although we cannot update the posted version at this time).
> > > > - y axis scaling in Fig. 1: We understand that using different scaling on the different axes makes them difficult to compare numerically, but this is not the purpose of the figure. Rather, the figure is intended to compare the shapes of the curves, i.e. that the first two curves (training loss and noisy validation loss) exhibit a drop around epoch 150, whereas the third curve (clean validation loss) does not drop at this point. This shape comparison would be more difficult to see if the axes all shared the same scaling.
> > > > - $\hat y_\lambda$: The reviewer is correct that this should be bold; thanks for noticing this. The notation has been corrected in the text (again, we are not able to update the posted version at this time).
> > > > - Reusing $i$ as both an index and the imaginary number: While we agree in general that notation reuse is to be avoided, in this case we do not believe another notation would be clearer. We only use $i$ as the imaginary number in a single equation in Sec. 3.2, with the description “where $i$ is the imaginary number” in the same sentence as the equation. Unfortunately, $i$ and $j$ are the only two letters commonly used to denote the imaginary number, and both are also the most commonly used letters to denote indices (along with $k$, which we use for image frequency, again a scenario in which it is a commonly-used letter). We are concerned that replacing any of these standard variable names would make things less clear, but are open to changing notation if the reviewer has a suggestion that would be more clear.
> > > >
> > > > Response to clarification questions:
> > > > 1. Fig. 1 shows only the target (“oracle”) functions (indeed the same ones used in Sec. 3.1), not the result of any network; we apologize for any miscommunication. All other figures in the paper show the results from networks, but Fig. 1 shows these oracle functions to validate our linear interpolation methodology and illustrate the connection between our two proposed measurement methods. As the oracle functions increase in frequency (the frequency of the label smoothing sinusoid), the proportion of high frequencies (as measured by linear interpolation) also increases.
> > > > 2. Perhaps we do not fully understand the reviewer’s question here. Fig. 2 is not showing results on any of our models, but rather introducing our linear interpolation method and validating it using the oracle functions used as training targets in our label smoothing method. Throughout our work, we consider a neural network as a function mapping an input image to a probability vector over classes, and are interested in the frequency decomposition of this function.
> > > > 3. We do not consider f=0.005 to be “high” in this work; the cutoff we use in quantifying the “high” frequencies in the linear interpolation experiment is f=0.05. We acknowledge that any cutoff would be somewhat arbitrary, but our results seem robust to a reasonable range of choices (roughly 0.03 to 0.15). We chose f=0.05 because it is 10% of the maximum frequency we can measure with our sampling distance of 1, and is near the upper end of the frequencies at which we observe interesting “dip” behavior with our label smoothing experiments. This choice is discussed in Sec. A.1.

---

> > > > > ### Comment · Reviewer_axgN · 2021-11-29
> > > > > **Comment**
> > > > >
> > > > > I am thankful to the authors for clarifying the responses. I believe in a future revision the notation should be improved (currently cluttered as mentioned in the previous comment), and also additional datasets should be included. This is important (as mentioned from the original review), since I am not confident many of the observations can be validated outside of CIFAR10 as mentioned [here](https://openreview.net/forum?id=e-IkMkna5uJ&noteId=93XTMClJElM). The latest author response mentions that no claim beyond CIFAR10 is made. However, I would argue that the title of the paper is `Spectral Bias in Practice: the Role of Function Frequency in Generalization', which it is even more general than image recognition. So, I would expect much more than even image recognition on various datasets. Similarly, nothing about CIFAR10 is mentioned in the abstract.
> > > > >
> > > > > Given the aforementioned facts, I will keep my score the same.

---

> > > > > > ### Author Response · Authors · 2021-11-29
> > > > > > **Update to abstract**
> > > > > >
> > > > > > Thank you for pointing out your concern about the specificity of claims on CIFAR10. Indeed we do not want our claims to be interpreted too broadly; we have updated our abstract to specifically state that our experiments use CIFAR10 (note that we cannot update the posted version at this time).

---

### Official Review · Reviewer_tm3W · 2021-10-30

**Correctness:** 3
**Technical Novelty And Significance:** 3
**Empirical Novelty And Significance:** 2
**Recommendation:** 8
**Confidence:** 4

**Main Review:**

Strengths:
Even though similar methodology has been used before (e.g. in Rahaman et al.), the proposed label smoothing technique seems to be novel in being applicable to multi-class classification. The experiments are extensive and well thought-out. The effects of training parameters on spectral bias (or its proxy) is investigated from various different directions. The paper is well-written and easy to follow. Overview of related work is sufficient.

Weaknesses:
It is not clear to me what tangible novel insights we learned about generalization beyond what has already been investigated and observed in the literature (such as Oymak et al. and similar works), namely that over-parameterized networks learn structure from data first, but can overfit to any noise over time. In other words, the paper makes observations on the effect of different regularization techniques on function frequencies but the *role* of function frequency in generalization is not investigated directly.
Moreover, the significance of the work is somewhat limited in my opinion because it is not clear how it can aid training choices. The key conclusion seems to be that 'an ideal function should include high enough frequencies to fit the data but avoid unnecessary high frequencies that can harm generalization', which is somewhat evident and it is not clear to me what consequences it has. For instance, can we use this methodology to inform us how to choose implicit/explicit regularization or network size?

Oymak, S., et al. "Generalization guarantees for neural networks via harnessing the low-rank structure of the jacobian." arXiv preprint arXiv:1906.05392 (2019).

Rahaman, N., et al. "On the spectral bias of neural networks." International Conference on Machine Learning. PMLR, 2019.

**Summary Of The Paper:**

This paper proposes a set of tools to probe spectral bias of deep neural networks, that is their tendency to learn low frequency, simpler functions earlier in training, whereas high frequencies are fit later. Authors propose adding noise to the labels through a target function via label smoothing, where the frequency and direction of the target function in image space can be varied. Moreover, they introduce a linear interpolation technique on validation data to probe the smoothness of the learned function along paths connecting natural images. Authors perform extensive experiments to demonstrate how the proposed tools can be used to investigate the spectral effect of training parameters such as model size and different forms of explicit and implicit regularization.

**Summary Of The Review:**

Overall, I think the paper has merit and the proposed tools have potential to reveal novel insights on generalization properties of deep neural networks. On the other hand, the core novelty and significance of the findings are somewhat limited in my opinion, therefore I recommend marginal acceptance. I am however open to change my score either way after hearing back from the authors and after further discussion with other reviewers.

---

> ### Author Response · Authors · 2021-11-12
> **Response to reviewer tm3W**
>
> We thank the reviewer for their time and feedback on our paper. Multiple reviewers suggested editing the presentation of our results to be quantitative rather than qualitative, and easier to interpret. Accordingly, in a separate comment for all reviewers, we provide aggregated metrics that summarize our findings quantitatively in additional figures. We will update the figures and text of the main paper as well to focus on clarity of presentation. We respond to reviewer-specific comments below.
> - We thank the reviewer for the reference [Oymak et al., 2019]; we will update our related work discussion accordingly. Indeed there has been much prior work, largely theoretical, on the spectral (and other) biases of neural networks, and this progress is what inspired our research. Our goal is to measure spectral bias in modern image classification networks and study empirically how different training decisions (like model size, regularization, and data augmentation) impact a model’s spectral bias. To the best of our knowledge, prior work has not included this sort of empirical analysis of spectral bias across many aspects of network training.
> - The reviewer makes an excellent point that the ultimate goal of our research is to improve our ability to design models with the “right” biases for specific tasks. Indeed it is our hope that our work will enable a more targeted approach to neural net “debugging”: if we can diagnose the root cause (in terms of spectral bias) of a network’s mistakes, we can then choose an appropriate intervention to improve performance. For example, if a model is very smooth in the regions around examples it misclassifies, perhaps data augmentation or increasing the model width (both of which increase the learned frequencies) will help. Likewise, if a model is very non-smooth around misclassified examples, perhaps regularization (which decreases the learned frequencies) will help. We plan to explore these applications of our work in the future; we note that such future research would not be possible without practical tools for measuring spectral bias, like those we propose in this work.
>
> Oymak, S., et al. "Generalization guarantees for neural networks via harnessing the low-rank structure of the jacobian." arXiv preprint arXiv:1906.05392 (2019).

---

> > ### Author Response · Authors · 2021-11-29
> > **Any other questions?**
> >
> > Dear reviewer tm3W, since the discussion period is drawing to a close, we wanted to check in and see if you have any further questions, or if our earlier response and revised manuscript satisfy your concerns. Thank you for your consideration.

---

> > > ### Comment · Reviewer_tm3W · 2021-11-29
> > > **Response to authors**
> > >
> > > I thank the authors for answering my concerns. Overall, I believe that the paper provides a useful tool that has the potential to improve our understanding of deep neural network generalization. I understand that the authors are going to address the practical deployment of the tool in future work. Given the extensive empirical study and the improvements made in updated version of the manuscript, I am going to increase my score to 7.

---

### Official Review · Reviewer_Vhe6 · 2021-11-02

**Correctness:** 2
**Technical Novelty And Significance:** 3
**Empirical Novelty And Significance:** 3
**Recommendation:** 8
**Confidence:** 5

**Main Review:**

## Strengths

1. **Research question**: I find the proposed research question of great interest to the community. The spectral bias of deep learning is mentioned very often as a justification of many practical tricks in the literature, but we are lacking a proper evaluation of this bias in practice.
2. **Analysis of many factors**: I admire the effort made by the authors in studying the effect that many factors of variability have in the spectral bias of SOTA convolutional neural networks. In particular, I find the study of the role of architecture and network capacity in the strength of the spectral bias very compelling.
3. **Connection between directional bias and spectral bias**: The directional inductive bias in the input space and the spectral bias in the function space are easily confused in the literature. In this sense, I find the analysis bridging the gap between these two inductive biases very interesting. In particular, I honestly believe that moving forward, the research community should strive to understand the interplay between the different types inductive biases present in neural networks, so I find that some of the observations in this work are a good step in the right direction.

## Weaknesses

1. **Evaluation metrics**: My main concern with this work is the fact that the choice of metrics used to evaluate the spectral bias of a neural network are highly subjective and difficult to interpret. Specifically, both the effective noise fitting curve and the cumulative difference norm in the linear path require to inspect a noisy curve to measure the spectral bias. This makes the final assessment of the results very qualitative. In my opinion, this seriously diminishes the strength of the presented results, and this has greatly affected my final score.

Nonetheless, I do not think this problem is unsolvable and I am willing to increase my score if this problem is addressed during the rebuttal. In particular, I would highly encourage the authors to look for alternative metrics to quantify the spectral bias, and perform some experiments to validate their qualitative results with a more numerically precise metric. As a suggestion, I propose the following two metrics:

   - **Test accuracy when training to predict sinusoidal labels of different frequency**: One way to precisely quantify the spectral bias would be by training multiple times on the same dataset with target labels of different frequency. In particular, I propose to generate a benchmark consisting on CIFAR10 samples and different target labels $S(X)$, or similar. The test performance obtained when training to predict different frequency targets should be a good indication of the spectral bias of that network.
   - **Roughness of loss landscape wrt the input**: One could use a similar metric as Mehmeti-Gopel et al. 2021 but in the input space to measure the frequency content of the final loss landscape of a trained neural network.
In any case, I encourage the authors to defend their choice of metric during the rebuttal if they believe there is no better metric to capture this phenomenon.

2. **Inconclusive results**: Another source of concern for me has been the fact that some of the presented analysis are rather speculative and not based on facts. In particular, when discussing the role of different types of regularization in the spectral bias, the authors refer multiple times to the connection between their observed spectral bias and the accuracy of the different models: "*This experiment reinforces the nuanced relationship between function complexity and performance...*", "*we find that [..] some smoothness [...] is beneficial, [but] too much causes detrimental underfitting*" or "*Neither complexity nor smoothness is, in itself, purely beneficial or detrimental: the trick is to balance them appropriately for the dataset.*" All these statements, however, are not supported by the data, as nowhere in this manuscript is the spectral bias of the different training runs compared to the performance of these models. Please, note that I am not implying that these statements are not true, I am just merely stating that these are not supported by the results presented in this manuscript. Reaching those conclusions could be possible if one juxtaposed some quantitative metric of the spectral bias to the final performance reached by these models, but I am afraid that with the current qualitative metrics, this is not possible.

## Other comments and additional questions
1. **Missing citations**: I believe some parts of this work could benefit from the inclusion of a few missing citations. In particular, prior to Ortiz-Jimenez et al. 2020, Yin et al. 2019 had also previously studied the robustness of computer vision classifiers to different frequency components. Also, on the implications of the spectral bias, Shah et al. 2020 argued that the simplicity bias was detrimental for robustness. The directional inductive bias presented in Sec. 4.2. was extensively studied in Ortiz-Jimenez et al. 2020. And finally, Mehmeti-Gopel et al. 2021 recently studied the roughness of the loss landscape of neural networks in an empirical campaign analogous to the one in parts of this work, although with respect to the weight space, rather than the input space.
   - Dong Yin, Raphael Gontijo Lopes, Jonathon Shlens, Ekin D. Cubuk, Justin Gilmer. "A Fourier Perspective on Model Robustness in Computer Vision". NeurIPS 2019
   - Harshay Shah, Kaustav Tamuly, Aditi Raghunathan, Prateek Jain and Praneeth Netrapalli. "The Pitfalls of Simplicity Bias in Neural Networks." NeurIPS 2020
   - Guillermo Ortiz-Jimenez, Apostolos Modas, Seyed-Mohsen Moosavi-Dezfooli and Pascal Frossard. "Neural Anisotropy Directions". NeurIPS 2020
   - Christian H.X. Ali Mehmeti-Göpel, David Hartmann and Michael Wand. "Ringing ReLUs: Harmonic Distortion Analysis of Nonlinear Feedforward Networks". ICLR 2021

2. **Possible methodological issues in linear interpolation**: I have two questions regarding the experiments measuring complexity of the loss landscape:
   - Why do the authors choose to work with segments of the same size, and do not fix the number of segments to query in the linear path? Do the results change significantly if one changes this element of the evaluation?
   - Doesn't the exclusion of the incorrectly labeled samples bias the final metric towards smoother values? Specifically, if one repeats this experiment without removing those samples, is the order of complexity of the different methods retained?

3. **Choice of architectures**: Could the authors elaborate on the reasons behind their choice of model architectures? Specifically, why did the authors decide to go with shake-shake models and very wide-resnets and did not provide also results for more manageable ResNets?
4. **Role of data distribution**: The authors blame the data distribution for the interplay between directional inductive bias and spectral bias in Sec. 4.2. However, this statement is also not supported by any evidence. In this sense, I wonder if the authors have tested whether the differences in the spectral bias for different directions are not present in isotropically distributed data, e.g., a normal distribution. If so, this statement would be better supported. Nevertheless, note that Ortiz-Jimenez et al. 2020 found that the directional inductive bias was mostly due to the flow of information in the architectures, so I would not expect that to be the case. In general, it would be good if the authors could comment on what do they expect if instead of using Fourier basis functions, they repeated their experiments using Neural Anisotropy Directions, instead.



**Summary Of The Paper:**

This work presents an empirical study of how different training aspects affect the spectral bias of neural networks in practice. To that end, the authors propose to inject label noise of different frequencies to the CIFAR10 dataset and suggest that the time a neural network takes to start overfitting this noise is a good metric of its spectral bias. They also propose to measure the variability of the loss landscape in the linear interpolation path between two images as proxy for spectral bias. Experiments on the effect of model architecture, explicit regularization, and data augmentation suggest that deep neural networks exhibit a strong spectral bias, in practice, which can be modulated by different design factors.

**Summary Of The Review:**

Although I believe the proposed research question is of great interest to the community, and I admire the effort made by the authors in studying the effect of many training factors in the variability of the spectral bias in CNNs; I think that the overreliance on qualitative metrics and lack of quantitative results makes most of the observations in this work inconclusive. I am open to increase my score if the authors address these issues during the rebuttal, but at this stage a lean towards rejection.

---

> ### Author Response · Authors · 2021-11-12
> **Response to reviewer Vhe6 (part 1 of 2)**
>
> We thank the reviewer for their time and feedback on our paper. Multiple reviewers suggested editing the presentation of our results to be quantitative rather than qualitative, and easier to interpret. Accordingly, in a separate comment for all reviewers, we provide aggregated metrics that summarize our findings quantitatively in additional figures. We will update the figures and text of the main paper as well to focus on clarity of presentation. We respond to reviewer-specific comments below.
> - Paper summary clarification: In the paper summary, the reviewer describes that we “propose to measure the variability of the loss landscape”, but this is not quite accurate. The loss landscape typically refers to the variation in the loss function (in our case, cross-entropy loss) as a function of the network parameters (model weights). However, our interpolation procedure studies the variation in the network predictions on individual images as a function of the input image.
> - Evaluation metrics: We thank the reviewer for their constructive suggestion about the clarity of our work. Please refer to the separate comment for all reviewers, in which we introduce quantitative summary metrics that we hope will clarify our results and make them easier for readers to interpret.
> - Other proposed metrics: Test accuracy when training to predict sinusoidal labels of different frequency–This proposal is quite similar to our label smoothing methodology, with two differences. Instead of just training on a target sinusoid, we use a target sinusoid as a label smoothing noise function so that the training target includes both the true class labeling and the target sinusoid. We believe that this is a more practically relevant way to measure spectral bias since it captures the spectral bias in the context of the real classification problem. The other difference is that we consider loss rather than accuracy, which is important because our label smoothing procedure does not change the maximum label (the true classification), so a model can fit our noise function without any loss in accuracy. Specifically, we consider the difference between clean and noisy test loss (rather than only considering performance on clean test labels), to ensure that our “effective noise fitting” metric captures the model actually fitting our target noise function and not just overfitting to the training data.
> - Other proposed metrics: Roughness of the loss landscape with respect to the input–We thank the reviewer for the reference and will update our related work discussion accordingly. As the reviewer points out, [Mehmeti-Gopel et al., 2021] considers frequency of the loss landscape, a function from model weights to loss, which is different from the frequency of the network, a function that maps from input image to class predictions, which we study. Both notions of frequency are interesting, but they are different. However, as the reviewer suggests it may be possible to adapt some of these methods to study function frequency; we will consider this in our future work.
> - Inconclusive results: We understand the reviewer’s concern about the clarity of the connection between the various training interventions we study and the final network performance. We will add the final network test accuracy values to the main text; for reference, they are (without data augmentation): 89.4% (wrn_32), 90.2% (wrn_160), 92.2% (shake_shake_32), 93.5% (shake_shake_96), 93.6% (shake_shake_112), and 95.8% (pyramid_net). However, we note that the claims we make in the paper rely only on the relative effect of each training intervention (whether it helps or hurts the final accuracy). In particular, we only consider interventions that boost final accuracy, namely increasing model width, modest weight decay, increasing the number of training examples, using data augmentation, and using self-distillation. Our claim stems from our observation that, among these beneficial interventions, some cause the model to be smoother (lower-frequency) and others cause the model to be more complex (higher-frequency).

---

> > ### Author Response · Authors · 2021-11-12
> > **Response to reviewer Vhe6 (part 2 of 2)**
> >
> > - Missing citations: We thank the reviewer for pointing us to further related work. We note that [Yin et al., 2019] is discussed in our related work section (in the paragraph titled “model sensitivity to image frequency”), as is a prior paper by Ortiz-Jimenez et al. (also from 2020). We will update our related work to include [Shah et al., 2020], [Ortiz-Jimenez et al., 2020b], and [Mehmeti-Gopel et al., 2021].
> > - Methodology of linear interpolation: The decision to use segments of the same size is relatively inconsequential; in fact we normalize our visualization so that we use a constant bin width for all paths regardless of length. That said, there are two minor reasons we chose to sample with constant sampling distance rather than a constant number of samples. The first is that by sampling at a constant, small interval and then averaging and normalizing within each lambda bin, we are able to measure with higher resolution and perhaps benefit from noise cancellation. The second is that with the same measurements of model prediction at regular intervals along each path, we can compute other proxy metrics of spectral bias, such as the discrete Fourier transform along each path. We considered several such metrics in our early experimentation but ultimately decided to present a simpler metric that is easier to understand (although we realize even the metric we use can be tricky to interpret).
> > - Exclusion of incorrectly labeled samples: We experimented with both of these options (excluding or including incorrectly labeled samples). Generally, the relative ordering (the qualitative interpretation of the result) is invariant with respect to this choice; we include examples for the weight decay experiment at: https://imgur.com/ZcKgGVl. Our decision to exclude incorrect examples in the paper was for aesthetic reasons, so that all curves would align at the bottom left of the plot even when comparing different models. However, if the reviewer feels it is preferable to include incorrect examples we can do so; this would not affect any of our qualitative results.
> > - Choice of architectures: We used the same architectures as in the AutoAugment paper [Cubuk et al., 2019a]. Our methodology should extend easily to any architecture; we used these models to be able to test the effects of data augmentation in a cohesive framework.
> > - Role of data distribution: We apologize for any miscommunication. We do not claim to have shown that the frequency distribution of natural images is the cause of the directional bias observed in Sec. 4.2, merely to point out that our finding agrees with the prediction of [Basri et al., 2020] so this is one possible explanation. It is also possible that this bias is inherent to the model architecture regardless of the data distribution, or that a combination of both effects is at play. Indeed, the Neural Anisotropy Directions discussed in [Ortiz-Jiminez et al., 2020b] are precisely the Fourier basis images we study. Considering these results jointly may help explain why CNNs perform well on natural image classification tasks, if their architecture induces a bias that is well-suited to the statistics of the data distribution, which reinforces that bias. This connection is an interesting direction for future research, and we will update the paper to include discussion of these multiple possible causes for the directional bias in Sec. 4.2.
> >
> >
> > Dong Yin, Raphael Gontijo Lopes, Jonathon Shlens, Ekin D. Cubuk, Justin Gilmer. "A Fourier Perspective on Model Robustness in Computer Vision". NeurIPS 2019
> >
> > Harshay Shah, Kaustav Tamuly, Aditi Raghunathan, Prateek Jain and Praneeth Netrapalli. "The Pitfalls of Simplicity Bias in Neural Networks." NeurIPS 2020
> >
> > Guillermo Ortiz-Jimenez, Apostolos Modas, Seyed-Mohsen Moosavi-Dezfooli and Pascal Frossard. "Neural Anisotropy Directions". NeurIPS 2020b
> >
> > Christian H.X. Ali Mehmeti-Göpel, David Hartmann and Michael Wand. "Ringing ReLUs: Harmonic Distortion Analysis of Nonlinear Feedforward Networks". ICLR 2021
> >
> > Ekin D. Cubuk, Barret Zoph, Dandelion Mane, Vijay Vasudevan, and Quoc V. Le.  Autoaugment:
> > Learning augmentation policies from data, 2019a.
> >
> > Ronen Basri, Meirav Galun, Amnon Geifman, David Jacobs, Yoni Kasten, and Shira Kritchman. Frequency bias in neural networks for input of non-uniform density, 2020.

---

> > > ### Comment · Reviewer_Vhe6 · 2021-11-19
> > > **Answer to authors (part 2 of 2)**
> > >
> > > > Methodology of linear interpolation [...]
> > >
> > > Thank you for the clarification. I think this point is now more clear. Nevertheless, since you also point out that you tried computing the DFT of the linear interpolation path, I wonder what make you decide not to provide this experiment in the paper. In general, even if it is in the Appendix, these additional views of the same result can be very valuable to the readers.
> > >
> > > > Exclusion of incorrectly labeled samples: [...]
> > >
> > > Thank you for the clarification. This is also much clearer.
> > >
> > > > Choice of architectures: [...]
> > >
> > > This is not the most standard practice in the literature, but this is just a minor comment.
> > >
> > > > Role of data distribution: [...]
> > >
> > > Finally, thank you again for the clarification. I also think that including this more nuanced discussion in the paper can help strengthen your message.

---

> > > > ### Author Response · Authors · 2021-11-22
> > > > **Response to reviewer**
> > > >
> > > > Thanks again for the feedback. We have introduced an FFT-based interpolation metric and updated the discussion of the image frequency experiment in the revised manuscript.

---

> > ### Comment · Reviewer_Vhe6 · 2021-11-19
> > **Answer to authors (part 1 of 2)**
> >
> > Thank you for your detailed response, and for [making the effort to improve the paper with quantitative metrics](https://openreview.net/forum?id=e-IkMkna5uJ&noteId=03xdGCRJkm). I will try to answer your comments one by one.
> >
> > > Paper summary clarification: In the paper summary, the reviewer describes that we “propose to measure the variability of the loss landscape”, but this is not quite accurate. [...]
> >
> > Please, note that even if the the loss landscape is  generally assumed to be a function of the weights, this is absolutely not always the case in the literature. For example, in the ample adversarial robustness literature the loss landscape is generally studied with respect to the input as this is the object of interest in that case. I appreciate the clarification by the authors, but I really believe I understood the paper correctly.
> >
> > > Other proposed metrics: Test accuracy when training to predict sinusoidal labels of different frequency–This proposal is quite similar to our label smoothing methodology, with two differences. [...]
> >
> > Thank you for defending your choice of metric. As I mentioned in my [answer to your general comment](https://openreview.net/forum?id=e-IkMkna5uJ&noteId=03xdGCRJkm), I believe that the new quantitative results are enough to support most of your claims in this regard. However, I also believe that this paper would benefit **also** from a complementary experiment where the authors provided the analogous results with just the sinusoidal labels (without the additive original labels). This would serve to isolate also the effect of the original label distribution from the actual results. In any case, this is just a personal opinion, and a minor comment at this point.
> >
> > > Other proposed metrics: Roughness of the loss landscape with respect to the input–We thank the reviewer for the reference and will update our related work discussion accordingly. [...]
> >
> > These are just suggestions, and by no means the only possibility. However, I would like to insist in saying that having a more direct way to measure the spectral components of the loss landscape (wrt input) rather than the "cumulative norm" would strengthen the paper. To the best of my knowledge, computing statistics over the FFT of the linear interpolation path should be straightforward.
> >
> > > Inconclusive results: We understand the reviewer’s concern about the clarity of the connection between the various training interventions we study and the final network performance. [...]
> >
> > Thank you very much for providing the additional accuracy numbers and clarifying your claims. However, even if I appreciate the relevance of these experiments, I still believe there is a certain degree of subjectivity in the way they are presented. In particular, I am concerned with the causal link you are trying to establish between function frequency and performance, *"Our experiments suggest that an ideal function should include high enough frequencies to fit the data but avoid unnecessary high frequencies that can harm generalization."*, when at most these results show that there is no clear correlation between function frequency and performance. I would therefore advise to tone down a bit these claims.

---

> > > ### Author Response · Authors · 2021-11-22
> > > **Response to reviewer**
> > >
> > > Thank you for the response and suggestions.
> > >
> > > Regarding the paper summary, we were not totally sure from the original review (since the term ``loss landscape'' can be overloaded) so we wanted to be completely clear in the rebuttal; it is good to know the reviewer indeed understood the paper correctly from the beginning.
> > >
> > > Regarding the suggestions of using a DFT-based metric to quantify the interpolation experiments, we agree this is desirable and have implemented the proposed change in the updated manuscript.
> > >
> > > We have also reworded these somewhat broad claims to be more precise and tied to our experiments.

---

### Official Review · Reviewer_9z2T · 2021-11-02

**Correctness:** 3
**Technical Novelty And Significance:** 2
**Empirical Novelty And Significance:** 3
**Recommendation:** 3
**Confidence:** 4

**Main Review:**

While the topic is certainly relevant and interesting, I think that the proposed methods are a bit hazy and I found some important missing details in the empirical exploration.

### Smoothness estimation methods:

1/ is a straight forward extension of Rahaman et al, applied to multiclass.
 - I don't really understand why you had to introduce "effective noise fitting" instead of just looking at the validation loss as in Rahaman et al?
 - What is the purpose of the 1/k added to the target vector ?
 - How did you choose the range of frequencies. E.g. in the exp in fig 3.a the frequency ranges from 0.035 to 0.04 which seems a rather tiny range compared to the experiments in Rahaman et al where the high frequency is 5x the low frequency
 - How can you tell from fig 3.a e.g. that the high frequencies are learned later than the low frequencies ? I see that the 0.035 yellow curve takes its minimum at around 50 epochs whereas the "high" frequency 0.04 dark blue one takes its minimum at around 30 epochs. Similarly in fig 3.b left where exactly is the minimum for wrn_160 ?

2/ is difficult to interpret precisely:
 - The role of delta is not defined, for instance what does it mean that "delta = 1" in section 3.2 ?
 - Why is it called *cumulative* difference norm? What is cumulative in this measurement?
 - In fig 2. if both images of the same class, why is ||y_lambda - y_0|| high at lambda=1 (i.e. exactly the other image)
 - How precisely do you see the smoothness in these plots? In some cases it is rather ambiguous e.g. in fig 6.c bottom which experiment in the range [0.1; 1.6] is the smoothest one?


### Empirical evaluation

 - The paper is lacking experimental details, such as choice of optimizer and hyperparameters. This is especially true as I would expect the learning rate to play an important role in how fast each frequency component is learned.
 - It would be interesting to include an ablation study when varying the learning rate.


You can also take inspiration from the method in [Zhang et al 2021] who study the spectral bias in the context of epoch-wise double descent, also using training examples, but by performing a full Fourier decomposition.

References:
Zhang, Xiong and Wu, Rethink the Connections among Generalization, Memorization, and the Spectral Bias of DNNs, IJCAI-21

**Summary Of The Paper:**

This paper proposes two methods for estimating smoothness of the prediction function learned by a deep network :
 1/ an extension of Rahaman et al. to multiclass classification, by adding a sine of desired frequency to the target vector, then training on the new target vectors. The "effective noise fitting" is then the difference between the validation loss on clean targets and the validation loss on modified targets.
 2/ a measure of change of the logits in l2 norm between 2 training examples, averaged over many examples

Using their methods, they show the effect on the smoothness of the predicted function of:
 - varying the number of parameters
 - adding explicit regularization (weight decay/mixup)
 - distillation

**Summary Of The Review:**

Overall, I find the paper in its current state not ready for publication. I think that the methodologies for measuring smoothness can be improved, and the experiments should include experimental details as well as an ablation study of the role of different hyperparameters on the smoothness of the function obtained after training.

---

> ### Author Response · Authors · 2021-11-12
> **Response to reviewer 9z2T**
>
> We thank the reviewer for their time and feedback on our paper. Multiple reviewers suggested editing the presentation of our results to be quantitative rather than qualitative, and easier to interpret. Accordingly, in a separate comment for all reviewers, we provide aggregated metrics that summarize our findings quantitatively in additional figures. We will update the figures and text of the main paper as well to focus on clarity of presentation. We respond to reviewer-specific comments below.
> - The purpose of effective noise fitting: While clean validation loss often shows the same qualitative behavior as effective noise fitting, we believe effective noise fitting is a more robust metric because it isolates the effects of the target noise function. Validation loss alone combines effects from the target noise function with effects from normal overfitting to the training data, which is separate from the one we want to measure.
> - The purpose of the 1/k: We leverage the standard formula for multiclass label smoothing, introduced in [Szegedy et al., 2015]. The 1/k term (should say S/k; typo will be corrected) is necessary in order to make the label vector a valid discrete probability distribution (all class probabilities sum to 1).
> - Choice of frequency range: We experimented with a wide range of frequencies, but for clarity presented a more restricted range where our models exhibit “dip” behavior. For frequencies below this range, the model fits the noise immediately. For frequencies above this range, the model never fits the noise during training. For reproducibility, we will update our appendix to include the full range of frequencies we tested for each model.
> - Interpreting Fig. 3a: We agree that these “dip” figures require some practice to read. We find that they are easiest to interpret by scanning from left to right until the curves separate, and then observing the relative ordering of the curves. For example, in Fig. 3a we see good separation between the curves around epoch 30, and in Fig. 3b the clearest separation is around epoch 60. However, we agree that this procedure is not ideal for ease of reading, so we propose a simpler metric and clearer style of figure in the separate comment for all reviewers.
> - The role of delta: In our interpolation experiments, delta determines which samples we measure along the interpolating path between two test images. Along this interpolating path, we evaluate the model at images that are spaced apart by image norm delta. We illustrate this in Fig. 2 (bottom), where delta is the distance (in image norm) between each of the images shown. Note that we use a much larger delta (i.e. fewer samples spaced farther apart) for illustration in Fig. 2, and a much smaller delta (i.e. more samples spaced closer together) in our actual interpolation experiments. The purpose of using a small delta is to ensure that we sample densely enough to see how quickly the model’s predictions change as we move through image space.
> - Cumulative difference norm: We apologize for any confusion caused by this naming and will add clarifications to our paper. By “cumulative” we mean relative to the first image along the interpolating path, rather than relative to the adjacent image along the interpolating path.
> - Interpreting interpolation figures: Similarly to the label smoothing figures, we decided based on reviewer feedback that a quantitative, summary metric makes our interpolation results easier to read. We provide updated figures in a separate comment for all reviewers.
> - Experimental details: We used the same learning rate schedule for all of our experiments; our training hyperparameters are based on those in [Cubuk et al., 2019a]. We will update the text to include all experimental details. Additionally, we provide an anonymized copy of our code as a supplemental file.
> - Ablation over learning rate: We thank the reviewer for the interesting suggestion; we will include a study of the effects of learning rate on spectral bias in the final version of our paper.
> - Method of [Zhang et al., 2021]: We thank the reviewer for pointing us to this relevant (and very recent) work. Indeed, this method bears some similarity to our interpolation method, but takes samples in a localized region around each image rather than sampling along interpolating paths. We believe that our method is complementary to [Zhang et al., 2021]; we offer a coarser but larger-scale measure of function smoothness whereas they offer a finer-grained but smaller-scale measure of function smoothness. We will update our discussion of related work accordingly.
> - Ablation over different hyperparameters: We will add an ablation study over learning rate. Our paper already includes studies of the effects of model architecture, model width, weight decay, data augmentation, training set size, and self-distillation; if the reviewer has specific suggestions for other parameters to study please let us know.

---

> > ### Author Response · Authors · 2021-11-12
> > **References for response to reviewer 9z2T**
> >
> > Christian Szegedy, Vincent Vanhoucke, Sergey Ioffe, Jonathon Shlens, and Zbigniew Wojna.  Re-thinking the inception architecture for computer vision, 2015.
> >
> > Ekin D. Cubuk, Barret Zoph, Dandelion Mane, Vijay Vasudevan, and Quoc V. Le.  Autoaugment:
> > Learning augmentation policies from data, 2019a.
> >
> > Xiao Zhang, Haoyi Xiong, and Dongrui Wu. Rethink the Connections among Generalization, Memorization, and the Spectral Bias of DNNs, 2021.

---

> > > ### Author Response · Authors · 2021-11-29
> > > **Any other questions?**
> > >
> > > Dear reviewer 9z2T, since the discussion period is drawing to a close, we wanted to check in and see if you have any further questions, or if our earlier response and revised manuscript satisfy your concerns. Thank you for your consideration.

---

### Author Response · Authors · 2021-11-12
**Response to all reviewers: clarity of results**

Multiple reviewers expressed concern that our figures represent results subjectively and require careful interpretation, making them difficult to understand in their current state. To address this concern, we propose augmenting our existing figures with “summary” figures that are both quantitative and easier to interpret.

Label smoothing experiments: We propose summarizing our “effective noise fitting” curves with the minimum value attained throughout the course of training. Visually, this corresponds to the “depth” of the “dip”. We find empirically that this is a reasonably robust metric for summarizing the degree to which different models fit the label smoothing noise, as it captures the maximum extent to which a network manages to fit the true labels relative to the injected noise. For example, we would summarize Fig. 3a (reproduced at https://imgur.com/VYh9XNX, with “effective noise fitting” renamed “noise fitting” for brevity) with the following: https://imgur.com/DnCxdEj.
As the frequency of the label smoothing noise increases (left to right on the x axis), the minimum noise fitting decreases, showing that models more readily fit lower-frequency noise (noise fitting is higher for low frequency noise).

We can use the same summary metric to clarify all of our label smoothing figures and results.

Model size (larger models fit noise more readily than smaller models): https://imgur.com/KhZ7nco

Sensitivity to Fourier image directions (models fit noise in low image frequency directions more readily than noise in high image frequency directions–note the exception for 0 image frequency, which the model is insensitive to because it is trained on normalized images): https://imgur.com/9DckdJU

Weight decay (models fit noise better when they are less strongly regularized): https://imgur.com/2t9b1Y5


A similar idea can also summarize our linear interpolation results, by considering the maximum value achieved by each curve across the full range of interpolation. For example, we propose summarizing Fig. 5b with https://imgur.com/e1EXIaH, in which it is clear that the model predictions have more variation when the model is less strongly regularized. The navy blue bars represent interpolation between examples from the same class, and the green bars represent interpolation between examples from different classes. From this updated figure we can also more readily observe that models are smoother (have less variation in their predictions) within-class compared to between-class.




Again, we can apply this updated visualization to all of our interpolation results.

Training set size (training with more examples produces a smoother function): https://imgur.com/ZtoU4k1

Data augmentation (RandAugment and AutoAugment produce slightly less smooth functions, whereas Mixup produces a much smoother function, compared to training without augmentation): https://imgur.com/ln4jhvP

Mixup strength (Increasing the Mixup strength makes models smoother between-class, but for large Mixup strength the model becomes less smooth within-class. We note that the Mixup paper [Zhang et al., 2018] recommends a moderate strength between 0.1 and 0.4, which in our experiment balances smoothness within-class and between-class): https://imgur.com/Kk012cK


Self-distillation (student model is smoother than teacher model): https://imgur.com/582xsbr


For reproducibility, we include an anonymized copy of our code (as a zip file).

Reviewer-specific questions and comments are addressed in individual responses. Please feel free to follow up if anything is still unclear. Thank you for your time and consideration.

Hongyi Zhang, Moustapha Cisse, Yann N. Dauphin, and David Lopez-Paz. Mixup: Beyond empirical risk minimization, 2018.

---

> ### Comment · Reviewer_axgN · 2021-11-18
> **No update in the manuscript?**
>
> The authors have replied to the questions, but it seems to me no changes have been made to the manuscript, while the response 'we will clarify our results' seems to be often used in the replies. Are the authors planning to update their manuscript in the rebuttal period?

---

> > ### Author Response · Authors · 2021-11-18
> > **In progress**
> >
> > We will post again when the updated manuscript is ready.

---

> > > ### Author Response · Authors · 2021-11-22
> > > **Updated manuscript**
> > >
> > > We have now updated the paper, including the following changes:
> > > - We have updated our label smoothing experiments to use the minimum noise fitting metric, as this is quantitative and easier to interpret visually.
> > > - We have updated our linear interpolation experiments to use a new metric based on the discrete Fourier transform of the neural net predictions along the sampled paths. We apply a threshold to the DFT frequencies and consider what fraction of the total magnitude is allocated to the high frequency components. This metric is precise in its frequency interpretation and easier to interpret visually, compared to our original figures.
> > > - We have updated our related work to include additional references, including those mentioned by the reviewers.
> > > - We have rewritten Section 3.2 (the methodology of linear interpolation) to more clearly describe our (updated) method, and updated Figure 2 and our description of it to directly connect our two experimental methods to each other and to the concept of function frequency.
> > > - We have updated our discussion of the image frequency experiment to include the possible contribution from the convolutional model architecture, as discussed in Ortiz-Jiminez et al., 2020.
> > > - We have reworded some of our claims to be more directly tied to our experimental findings.
> > > - We have updated all of our interpolation experiments to include all image pairs, rather than excluding pairs where either image is misclassified. This is simpler and does not affect our results in any meaningful way.
> > > - We have included additional experimental details in Section 3.3, including optimizer and learning rate schedule as well as the clean test accuracies of our models. Our full codebase is available in the supplement. Additional details for the updated interpolation method are included in the updated Appendix.
> > >
> > > We thank the reviewers for their constructive suggestions; we believe the updated paper is clearer than the original version. Further feedback is also welcome.

---

> ### Comment · Reviewer_Vhe6 · 2021-11-19
> **Step in the right direction**
>
> Thank you very much for making the effort to quantify your previously qualitative results. Having looked at your new plots, I honestly believe this is a step in the right direction. Indeed using the minimum *noise fitting* value seems to adequately capture most of the observed phenomena. I am less convinced by the linear interpolation summaries. The numbers seem to be consisent with what previously observed, but the interpretation in terms of frequency is very indirect. What is the added benefit of the proposed metric over computing statistics directly over the FFT of the functions in the linear interpolation paths?
>
> As [Reviewer axgN](https://openreview.net/forum?id=e-IkMkna5uJ&noteId=fh_ZLOPAz-z), I would also encourage the authors to update their manuscript during the rebuttal period. At the moment, I am willing to increase my score to 6, based on the promised changes, but I would prefer to see those changes implemented before raising my score. The reasons why I do not increase more my score are mostly that the second quantitative metric does not convince me (see above) and that [the causal link between frequency and performance established by the authors is only tenously substantiated](https://openreview.net/forum?id=e-IkMkna5uJ&noteId=an5A8ZtydbH). Further discussions and changes in this regard might make me increase more on my score.

---

> > ### Author Response · Authors · 2021-11-22
> > **Updated manuscript**
> >
> > Thank you for your constructive suggestions. We have updated the paper to include a quantitative, DFT-based metric to summarize our interpolation experiments, as well as reworded some of our claims to be more directly tied to our experimental results. Please take a look and let us know if you have further feedback.

---

> > > ### Comment · Reviewer_Vhe6 · 2021-11-23
> > > **Increase in score**
> > >
> > > Thank you for making the effort to update your manuscript. I honestly believe these changes have made your submission stronger. In this sense, having read the other reviewer comments, the author's responses, and the new version of the manuscript, I have decided to increase my final score to an 8: accept, good paper.
> > >
> > > I still believe the paper can be significantly improved, mostly in terms of writing and presentation of the results; but also by exploring alternative techniques to study the frequency content and spectral bias of deep neural networks, and investigating results on other datasets, as suggested by other reviewers. This being said, I also feel that the presented results are valuable for the community and technically sound, so rejecting this paper based on these issues would be, in my opinion, too severe; especially considering the additional toll this would impose on the authors.

---

### Decision · Program_Chairs · 2022-01-20

**Decision:**

Reject

**Comment:**

This paper expands the spectral bias, which has been studied in a constrained situation such as the fully-connected network, to a more practical situation of a multi-class classification situation, and proposes a novel technique that can measure the smoothness through linear interpolation of test examples.

Two reviewers highly evaluated the importance of the research question considered in this study and the value of diverse experiments applying the proposed method in various directions, and suggested acceptance. On the other hand, two other reviewers suggested rejection due to the lack of rigor in writing and experiments. I strongly agree with the reviewer's concern that  the method was only verified  on CIFAR10 and the rigor of the experiment was lacking. Unlike the spectral bias paper, which is the basis of this study, this submission is not a theoretical paper, but rather an experimental paper. I admit that it is impossible to verify in various domains as mentioned by the author. However, I believe that verification on more diverse, especially larger-scale datasets is essential at least focusing on the image classification task.